# CONTRASTIVE DIFFUSION GUIDANCE FOR SPATIAL INVERSE PROBLEMS

**Sattwik Basu**[1]*      **Chaitanya Amballa**[1]*      **Zhongweiyang Xu**[1]      **Jorge Vančo Sampedro**[1]
**Srihari Nelakuditi**[2]      **Romit Roy Choudhury**[1]

[1]University of Illinois Urbana-Champaign     [2]University of South Carolina

## ABSTRACT

We consider a class of inverse problems characterized by forward operators that are partially specified, non-smooth, and non-differentiable. Although generative inverse solvers have made significant progress, we find that these forward operators introduce a distinct set of challenges. As a concrete instance, we consider the problem of reconstructing spatial layouts, such as floorplans, from human movement trajectories, where the underlying path-generation process is inherently non-differentiable and only partially known. In such problems, direct likelihood-based guidance becomes unstable, since the underlying path-planning process does not provide reliable gradients. We break-away from existing diffusion-based posterior samplers and reformulate likelihood-based guidance in a smoother embedding space. This embedding space is learned using a *contrastive* objective to bring compatible trajectory-floorplan pairs close together while pushing mismatched pairs apart. We show that this surrogate likelihood score in the embedding space provides a valid approximation to the true likelihood score, making it possible to steer the denoising process towards the posterior. Across extensive experiments, our model **CoGuide** produces more consistent reconstructions and is more robust than existing inverse-solvers and guided diffusion. Beyond spatial mapping, we show that our method can be applied more broadly, suggesting a route toward solving generalized *blind* inverse problems using diffusion models.

## 1 INTRODUCTION

Inverse problems (IP) seek to recover unknown signals from indirect, partial, and often noisy measurements. The unknown signal $\mathbf{x} \in \mathbb{R}^m$ and the measurement $\mathbf{y} \in \mathbb{R}^l$ are related via a forward process $\mathbf{y} = \mathcal{A}(\mathbf{x}, \mathbf{n})$, where $\mathcal{A}: \mathbb{R}^m \to \mathbb{R}^l$ is a forward operator and $\mathbf{n}$ denotes the measurement noise. The objective is to estimate $\mathbf{x}$ when given only the observation $\mathbf{y}$, that is, to construct a suitable inverse map $\mathcal{A}^\dagger$ such that $\mathbf{x} \leftarrow \mathcal{A}^\dagger(\mathbf{y})$.

The fundamental challenge in inverse problems is ill-posedness Hadamard & Morse (1953) which necessitates the use of structural priors. Past work has made remarkable progress using insightful observations on the nature of $\mathbf{x}$, leading to hand-crafted priors like sparsity, total variation, etc. Engl et al. (1996). These priors make optimization tractable (e.g., maximum a posteriori) but often underfit complex structure and need careful tuning to balance fidelity and regularization. In recent years, diffusion models Sohl-Dickstein et al. (2015); Song et al. (2020b); Ho et al. (2020) have become a powerful line of attack since they can extract priors from large datasets, and use the prior to sample from the posterior distribution Zheng et al. (2025); Chung et al. (2022); Song et al. (2023). Active progress is being made along the axis of operator complexity—starting from linear and non-linear operators, and going into non-differentiable, partially observable, and even blind functions. *This paper* brings forth a reasonably challenging (path-planning) operator, motivated by a practical application. Let us present the application first and then shed light on the operator.

Consider a user walking around in her home for a few minutes. Using some sensor, e.g., a smartphone, the user's trajectory has been recorded. This trajectory is a sequence of location measurements inside the home, $\mathbf{y} = [y_1, y_2, ... y_n]$, along which the user has walked. *We ask*, given this trajectory measurement, is it possible to infer the floorplan $\mathbf{x}$ of the home, where floorplan is the dimensions and layouts of the walls in the home. Observe that this is a *spatial inverse problem*, modeled as $\mathbf{y} = \mathcal{A}(\mathbf{x}, \mathbf{n})$, because the way the human user walks, $\mathbf{y}$, is indeed a function of the layout of the home $\mathbf{x}$. This function is the $\mathcal{A}(.)$ operator, a policy in the user's brain that plans the path from point A to point B, for a given floorplan

---

*Equal contribution.

$\mathbf{x}$. This path-planner is complex—it models factors such as distance walked, collision with walls and furniture, time to walk, number of turns, etc. Optimizations with such functions inherit this complexity since tiny changes in the floorplan—say a small hole in one wall—can drastically change the planned path. Hence, $\mathcal{A}(.)$ is in the regime of non-linear, non-differentiable, and partially observable operators, presenting a relatively new question (to the best of our knowledge) to diffusion-based inverse solvers. This paper concentrates on this specific problem of inverse floorplan estimation, but shows the potential to extend the approach to a broader family of spatial inverse problems.

Our key idea follows score-based posterior sampling in which the posterior score $\nabla_{\mathbf{x}}\log p(\mathbf{x}|\mathbf{y})$ decomposes into a diffusion-learned prior score $\nabla_{\mathbf{x}}\log p(\mathbf{x})$ and a likelihood score $\nabla_{\mathbf{x}}\log p(\mathbf{y}|\mathbf{x})$ Song et al. (2020b). The prior term injects structural knowledge about plausible $\mathbf{x}$, while the likelihood term enforces consistency with the measurement $\mathbf{y}$. Following Diffusion Posterior Sampling (DPS) Chung et al. (2022), we approximate the likelihood score as $\nabla_{\mathbf{x}}\log p(\mathbf{y}|\hat{\mathbf{x}}_0)$ using Tweedie's mean estimate $\hat{\mathbf{x}}_0=\mathbb{E}[\mathbf{x}_0|\mathbf{x}_t]$ Efron (2011). When $\mathcal{A}(.)$ is known and differentiable with additive $\mathbf{n}\sim\mathcal{N}(\mathbf{0},\sigma^2\mathbf{I})$, the likelihood score reduces to $\nabla_{\mathbf{x}}\|\mathbf{y}-\mathcal{A}(\hat{\mathbf{x}}_0)\|_2^2$ and provides guidance for the denoising process in diffusion. In our case, the path planning operator $\mathcal{A}(.)$ is difficult to model and non-smooth, and as we show, various approximations of $\mathcal{A}(.)$—even when differentiable—produce poor results due to the instability in optimization.

In light of this, we break-away from convention and project both, floorplans $\mathbf{x}$ and trajectory $\mathbf{y}$, into a common embedding space $\mathcal{E}$, in which the likelihood score assumes a surrogate form:
$$\nabla_{\mathbf{x}}\|[\hat{\mathbf{x}}_0]_{\mathcal{E}}-[\mathbf{y}]_{\mathcal{E}}\|_2^2 \tag{1}$$
where $[.]_{\mathcal{E}}\in\mathcal{E}$. We train this embedding space using a contrastive approach Jaiswal et al. (2021); Le-Khac et al. (2020) that pulls matching $\langle$trajectory, floorplan$\rangle$ pairs closer to each other, and pushes away pairs that are incompatible. In other words, the embedding space implicitly learns the $\mathcal{A}(.)$ operator from matching pairs of floorplan and trajectory data, where the latter is synthetically generated from the former using an approximate $\mathcal{A}(.)$ operator. Importantly, the likelihood term in the embedding space is a smoother function for optimization and we show that it is a valid surrogate of the original intractable likelihood score.

We train our Diffusion prior using public floorplan datasets; during inference, our method **CoGuide** generates floorplans for given (sparse, medium, or dense) trajectories. Results reliably outperform 6 different baselines: 3 that are augmentations of DPS with path-planners Yonetani et al. (2021); Kirilenko et al. (2023); Liu et al. (2024), 2 that are established inverse solvers Zhu et al. (2023); Wang et al. (2024), and 1 classifier free guidance (CFG) diffusion model Ho & Salimans (2022) that (over)fits to the joint distribution of trajectories and floorplans. We believe **CoGuide** has potential beyond this specific application of floorplan inference and demonstrate generalization through blind audio restoration (Appendix D), an inverse problem where the forward degradation operator is completely unknown. We discuss early thoughts on follow-on research directions to leverage contrastive learning in diffusion-based inverse solvers.

## 2 PRELIMINARIES

**Diffusion models** Sohl-Dickstein et al. (2015); Ho et al. (2020); Song et al. (2020a;b) are a class of generative models capable of producing high-quality samples across a wide range of domains, including images Dhariwal & Nichol (2021), audio Kong et al. (2020b); Liu et al. (2023), video Ho et al. (2022), and 3D data Luo & Hu (2021); Poole et al. (2022). These models define generation as the reversal of a forward noising process that is formalized using a stochastic differential equation (SDE).
$$d\mathbf{x}_t=f(\mathbf{x}_t,t)dt+g(t)d\mathbf{w}_t, \tag{2}$$
where $\mathbf{x}_t\in\mathbb{R}^d$ is the state at time $t\in[0,T]$, $f(\mathbf{x}_t,t)$ is the drift, $g(t)$ is the diffusion coefficient, and $\mathbf{w}_t$ is a standard Brownian motion (Wiener process). Starting from clean data $\mathbf{x}_0\sim p_{\text{data}}$, this forward process gradually corrupts the signal so that $\mathbf{x}_T\sim\mathcal{N}(0,\mathbf{I})$. The associated reverse-time SDE, derived by Anderson (1982), recovers data from noise:
$$d\mathbf{x}_t=[f(\mathbf{x}_t,t)-g(t)^2\nabla_{\mathbf{x}_t}\log p_t(\mathbf{x}_t)]dt+g(t)d\tilde{\mathbf{w}}_t, \tag{3}$$
where $p_t(\mathbf{x}_t)$ is the marginal density of $\mathbf{x}_t$ at time $t$, $\nabla_{\mathbf{x}_t}\log p_t(\mathbf{x}_t)$ is the (time-dependent) score function, and $\tilde{\mathbf{w}}_t$ is reverse-time Brownian motion.

**Inverse Solvers.** Diffusion models have also been adapted for inverse problems based on the insight that one can design a reverse SDE using the posterior score as:
$$\underbrace{\nabla_{\mathbf{x}_t}\log p_t(\mathbf{x}_t|\mathbf{y})}_{\text{posterior score}}=\underbrace{\nabla_{\mathbf{x}_t}\log p_t(\mathbf{x}_t)}_{\text{prior score}}+\underbrace{\nabla_{\mathbf{x}_t}\log p_t(\mathbf{y}|\mathbf{x}_t)}_{\text{likelihood score}}. \tag{4}$$

Here, the prior score is easy to approximate by a trained diffusion model $s_\theta(\mathbf{x}_t,t)$. The likelihood score, in contrast, is intractable as $\mathbf{y}$ depends only on $\mathbf{x}_0$, not directly on $\mathbf{x}_t$. In response to this, prior work such as Diffusion Posterior Sampling (DPS) Chung et al. (2022) have approximated the likelihood term as $p_t(\mathbf{y}|\mathbf{x}_t) \approx p(\mathbf{y}|\hat{\mathbf{x}}_0)$ where $\hat{\mathbf{x}}_0 := \hat{\mathbf{x}}_0(\mathbf{x}_t) = \mathbb{E}[\mathbf{x}_0|\mathbf{x}_t]$ is obtained from a single denoising step using Tweedie's formula: $\mathbb{E}[\mathbf{x}_0|\mathbf{x}_t] = \mathbf{x}_t + \sigma_t^2 \nabla_{\mathbf{x}_t} \log p_t(\mathbf{x}_t)$

This formulation has been successfully used to guide diffusion models in solving a variety of inverse problems across scientific domains Zheng et al. (2025). **CoGuide** builds on this framework but makes a necessary departure owing to the challenges posed by the path-planning operator. We briefly discuss these operators before formulating the floorplan inference inverse problem.

**Path Planners as Forward Operators.** Unlike typical inverse problems (e.g., deblurring or inpainting), our measurements $\mathbf{y}$ are human-walked trajectories; thus, the forward operator $\mathcal{A}$ encodes a walking policy through an indoor layout (the floorplan $\mathbf{x}$). Directly modeling human navigation is difficult, so we use path-planners as proxies LaValle (2006); Hart et al. (1968); Gammell et al. (2015); Noreen et al. (2016). Empirical evidence indicates people favor short, direct routes with few turns Tong & Bode (2022) which aligns well with shortest-path planning. Therefore, the classical A* algorithm Hart et al. (1968) is a good choice for the forward operator. The floorplan is discretized into a grid graph, and given a start and end location, A* computes the shortest collision-free path, which we take as the predicted walking trajectory. To fit into the inverse problem framework, differentiable variants of A*, such as Neural A* Yonetani et al. (2021), TransPath Kirilenko et al. (2023), NRRT Wang et al. (2020), and Takahashi et al. (2019) are of interest; they all aim to enable gradient-based learning. Recent diffusion-based planners (DiPPeR Liu et al. (2024), PbDiff Luo et al. (2024)) further provide differentiable path generation. When inserted as forward operators, these planners often induce non-smooth objectives where small layout perturbations can cause large path changes, which in turn hinders convergence in gradient-based optimization. We analyze causes for this instability next and motivate our embedding-space likelihood surrogate.

## 3 METHOD

### 3.1 PROBLEM FORMULATION

Fig. 1 (Left) top row shows an unknown 2D floorplan $\mathbf{x} \in \mathbb{R}^{m \times n}$. A user walks in this floorplan (from point A to a destination point B, then from B to another destination C, and so on) and records a sequence of location measurements. The union of all these location measurements on a 2D image gives us the trajectories $\mathbf{y} \in \mathbb{R}^{m \times n}$ as shown in Fig. 1 (Left) bottom row. Given location sensors are noisy, the forward process is $\mathbf{y} = \mathcal{A}(\mathbf{x}) + \mathbf{n}$ where $\mathcal{A} : \mathbb{R}^{m \times n} \to \mathbb{R}^{m \times n}$ approximates the human walk using a (generally nonlinear) A* path-planner, and $\mathbf{n}$ models additive Gaussian noise of the location sensor.

**Our goal** is to utilize the DPS framework (Eq. 5 below) for which we need the likelihood score.

$$\nabla_{\mathbf{x}_t} \log p_t(\mathbf{x}_t|\mathbf{y}) \approx s_\theta(\mathbf{x}_t,t) + \nabla_{\mathbf{x}_t} \log p_t(\mathbf{y}|\hat{\mathbf{x}}_0) \tag{5}$$

Approximating the likelihood requires propagating $\hat{\mathbf{x}}_0$ through the $\mathcal{A}(.)$ operator and using its gradient to steer the diffusion prior $s_\theta(\mathbf{x}_t,t)$; doing this stably lies at the heart of our problem.

The stability issues are due to a number of factors, partly depending on the realization of the $\mathcal{A}(.)$ operator. Observe that a path planning algorithm must perform local searches at every intermediate point while growing a path from the source to the destination. The path grows to a new pixel when that pixel index *minimizes* the path cost towards the destination; this $\mathrm{argmin}$ operation makes the process non-differentiable. Differentiable approximations such as Neural A* (NA*) Yonetani et al. (2021), Transpath Kirilenko et al. (2023), and DiPPeR Liu et al. (2024), mitigate the pixel selection problem, however, the Jacobian derived from the likelihood score proves to be very sensitive. Said differently, the likelihood score $\nabla_{\mathbf{x}} \|\mathbf{y} - \mathcal{A}(\mathbf{x})\|_2^2 = -2 J_{\mathcal{A}}(\mathbf{x})^\top (\mathbf{y} - \mathcal{A}(\mathbf{x}))$ contains the Jacobian $J_{\mathcal{A}}(\mathbf{x})$ and $\|J_{\mathcal{A}}(\mathbf{x})\|$ is large. Intuitively, this happens because the path chosen by the planner is immune to most pixels in the floorplan; however, if a few pixels change slightly, then the new chosen path can be dramatically different.

To visualize this effect, the top row of Fig. 1 (Left) shows paths from different path planners on a floorplan with fixed start and end locations (note that NA* outputs all the visited pixels called histories to remain differentiable). The bottom row is a slightly different floorplan where three small doors have been introduced in each of the lower vertical walls. These door pixels prompted the planners to significantly change their paths, indicating a highly non-smooth $\mathcal{A}(\cdot)$. Steering the diffusion prior $s_\theta(\mathbf{x}_t,t)$ with such a non-smooth guidance from the $\nabla_{\mathbf{x}_t} \log p_t(\mathbf{y}|\hat{\mathbf{x}}_0)$ is unstable. Lastly, these planners are generally trained

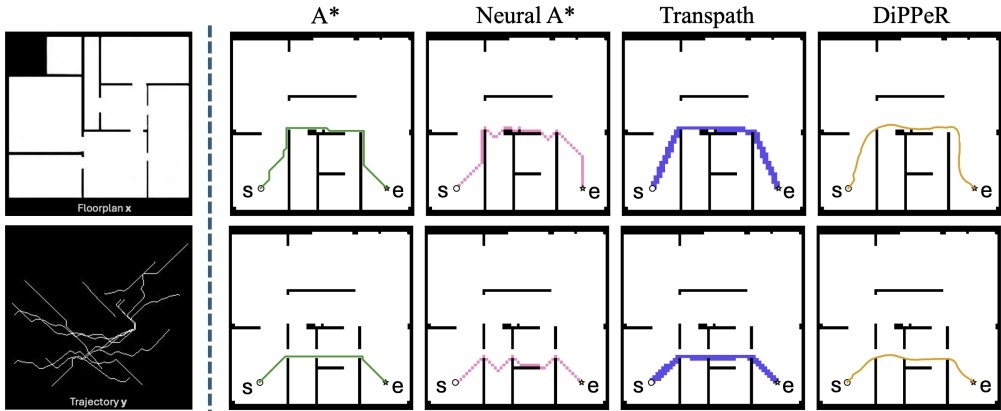

Figure 1: **(Left)** Example floorplan **x** and measured human-walked trajectory **y**. **(Right)** Paths chosen by A*, Neural A*, TransPath, and DiPPeR from the same start and end locations ("s" and "e"). The bottom row is a slight change from the top row, prompting a large change in path selection.

on binary floorplans (black walls and white empty space) and must cope with continuous-valued inputs (gray pixels) during the reverse diffusion process.

## 3.2 Guidance through Contrastive Learning

Given the complexity in harnessing differentiable path-planning operators, we side-step the issue entirely. Instead, we propose to design a surrogate for the likelihood score in a learned embedding space $\mathcal{E} \subset \mathbb{R}^d$ that is **smooth** (Lipschitz, without discontinuities) and **aligned** (compatible floorplan–trajectory pairs map nearby, mismatched pairs far apart). We expect that operating in such a space will stabilize the gradients of the likelihood surrogate, making the guidance to the denoiser smoother. Of course, we need to ensure that this new likelihood score from the embedding space is still a valid approximation for the original likelihood score $\nabla_{\mathbf{x}_t} \log p_t(\mathbf{y}|\hat{\mathbf{x}}_0)$.

We construct the space $\mathcal{E}$ using two encoders $f_\varphi$ (for floorplans), and $g_\psi$ (for trajectories):

$$f_\varphi \colon \mathcal{X} \to \mathcal{E}, \quad \mathbf{x} \mapsto [\mathbf{x}]_\mathcal{E} = f_\varphi(\mathbf{x}), \qquad g_\psi \colon \mathcal{Y} \to \mathcal{E}, \quad \mathbf{y} \mapsto [\mathbf{y}]_\mathcal{E} = g_\psi(\mathbf{y}). \tag{6}$$

where $\mathbf{x} \in \mathcal{X}$ and $\mathbf{y} \in \mathcal{Y}$ such that $\|f_\varphi(\mathbf{x})\|_2 = \|g_\psi(\mathbf{y})\|_2 = 1$. We then use these encoders to define the likelihood-to-evidence ratio for a pair of inputs $(\mathbf{x}, \mathbf{y})$ in the form of an un-normalized distribution $\pi$:

$$\pi(\mathbf{y}, \mathbf{x}) \propto \exp(\langle f_\varphi(\mathbf{x}), g_\psi(\mathbf{y}) \rangle / \tau) \tag{7}$$

where, $\langle .,. \rangle$ denotes an inner product, and temperature $\tau > 0$ controls the concentration of this distribution on the unit-hypersphere in $\mathbb{R}^d$. *Intuitively*, when this embedding space is learned correctly, larger inner products should correspond to higher pairwise compatibility and thus higher likelihood.

To achieve this, we train $f_\varphi$ and $g_\psi$ *contrastively* using an InfoNCE-style loss function Le-Khac et al. (2020); Oord et al. (2018). This approach naturally organizes the embedding space by pulling *matched* pairs together while pushing *unmatched* pairs apart Wang & Isola (2020).

**Contrastive similarity as a likelihood surrogate.** To see why this is a valid approach to approximating the true likelihood, we note that InfoNCE links contrastive learning to density estimation. Following Oord et al. (2018) we observe that the optimal contrastive classifier recovers a likelihood-to-evidence ratio. That is, when the InfoNCE loss attains its optimum, we get

$$\exp(\langle f_\varphi(\mathbf{x}), g_\psi(\mathbf{y}) \rangle / \tau) \propto \frac{p(\mathbf{y}|\mathbf{x})}{p(\mathbf{y})} \implies \frac{1}{\tau} \langle f_\varphi(\mathbf{x}), g_\psi(\mathbf{y}) \rangle = \log p(\mathbf{y}|\mathbf{x}) - \log p(\mathbf{y}) + C \tag{8}$$

where $C$ is a constant independent of **x**. Taking gradients with respect to **x** in Eq. 8 exactly recovers the likelihood score on the right-hand side i.e., $\frac{1}{\tau} \nabla_x \langle f_\varphi(\mathbf{x}), g_\psi(\mathbf{y}) \rangle = \nabla_\mathbf{x} \log p(\mathbf{y}|\mathbf{x})$. Therefore, we substitute this surrogate likelihood into Eq. 5) along with the DPS approximation which gives,

$$\nabla_{\mathbf{x}_t} \log p_t(\mathbf{x}_t|\mathbf{y}) \approx s_\theta(\mathbf{x}_t, t) + \frac{1}{\tau} \nabla_{\mathbf{x}_t} \langle f_\varphi(\hat{\mathbf{x}}_0(\mathbf{x}_t)), g_\psi(\mathbf{y}) \rangle$$

$$= s_\theta(\mathbf{x}_t, t) - \frac{1}{2\tau} \nabla_{\mathbf{x}_t} \|f_\varphi(\hat{\mathbf{x}}_0(\mathbf{x}_t)) - g_\psi(\mathbf{y})\|_2^2. \tag{9}$$

The second equality is valid since unit-norm embeddings satisfy $\langle \mathbf{u}, \mathbf{v} \rangle = 1 - \frac{1}{2} \|\mathbf{u} - \mathbf{v}\|_2^2$. Therefore, the contrastive guidance admits an equivalent squared-distance form. On the whole, since $f_\varphi(\cdot)$ and $g_\psi(\cdot)$ are smooth, the resulting gradients of $\|f_\varphi(\hat{\mathbf{x}}_0(\mathbf{x}_t)) - g_\psi(\mathbf{y})\|_2^2$ are stable, steadily steering the reverse diffusion toward floorplans whose embeddings are compatible with the measured trajectory. We extend this connection from InfoNCE to supervised, multi-positive contrastive learning Khosla et al. (2020); this is natural in our setting since several compatible trajectories can be synthesized from a single floorplan. This modification maintains the validity of the likelihood surrogate during inference as explained in Appendix A.

**Contrastive Loss Functions.** We train the encoders $f_\varphi$ and $g_\psi$ with a symmetric supervised contrastive objective. In this setting, let $p^+(\mathbf{x}, \mathbf{y})$ denote the distribution of matched (positive) floorplan–trajectory pairs, and $p(\mathbf{x})$, $p(\mathbf{y})$ be the marginals used to draw negatives. The expectations below are estimated with in-batch negatives (and multi-positives when available).

$$\mathcal{L}_{f \to t} = -\mathbb{E}_{(\mathbf{x},\mathbf{y}) \sim p^+} \log \frac{\exp\big(\langle f_\varphi(\mathbf{x}), g_\psi(\mathbf{y}) \rangle / \tau\big)}{\exp\big(\langle f_\varphi(\mathbf{x}), g_\psi(\mathbf{y}) \rangle / \tau\big) + \mathbb{E}_{\mathbf{y}^- \sim p(\mathbf{y})}\big[\exp\big(\langle f_\varphi(\mathbf{x}), g_\psi(\mathbf{y}^-) \rangle / \tau\big)\big]} \tag{10}$$

Here the floorplan $\mathbf{x}$ acts as the anchor, and the objective pulls the matching trajectory $\mathbf{y}$ close to $f_\varphi(\mathbf{x})$ while pushing away non-matching trajectories $\mathbf{y}^- \sim p(\mathbf{y})$. This aligns trajectories around the correct floorplan anchor and shapes a locally smooth neighborhood in $\mathcal{E}$.

Conversely, we anchor on the trajectory $\mathbf{y}$ and attract the matching floorplan $\mathbf{x}$ while repelling non-matching floorplans $\mathbf{x}^- \sim p(\mathbf{x})$. This complements the floorplan-anchored view and provides bi-directional consistency of the embedding space

$$\mathcal{L}_{t \to f} = -\mathbb{E}_{(\mathbf{x},\mathbf{y}) \sim p^+} \log \frac{\exp\big(\langle g_\psi(\mathbf{y}), f_\varphi(\mathbf{x}) \rangle / \tau\big)}{\exp\big(\langle g_\psi(\mathbf{y}), f_\varphi(\mathbf{x}) \rangle / \tau\big) + \mathbb{E}_{\mathbf{x}^- \sim p(\mathbf{x})}\big[\exp\big(\langle g_\psi(\mathbf{y}), f_\varphi(\mathbf{x}^-) \rangle / \tau\big)\big]} \tag{11}$$

**Adding Alignment Losses.** To improve performance, we train the contrastive model with the symmetric losses in Eq. 10, 11 along with an additional alignment loss that further pulls each matched floorplan–trajectory pair closer in the embedding space.

$$\mathcal{L}_{\text{align}} = \mathbb{E}_{(\mathbf{x},\mathbf{y}) \sim p^+} \|g_\psi(\mathbf{y}) - f_\varphi(\mathbf{x})\|_2^2 \tag{12}$$

While the loss $\mathcal{L}_{\text{contra}}$ separates positives from in-batch negatives, the alignment term primarily tightens each true pair by shrinking the intra-positive $L_2$ distance. To avoid hindering learning in early epochs, we start with only $\mathcal{L}_{f \to t}$ and $\mathcal{L}_{t \to f}$ and slowly increase the alignment weight after a few epochs. This schedule gives cleaner clusters in the embedding space and better results in practice. **CoGuide**'s final contrastive loss function can now be expressed as:

$$\mathcal{L}_{\text{contra}} = \lambda \mathcal{L}_{f \to t} + (1 - \lambda)\mathcal{L}_{t \to f} + \lambda_{align}\mathcal{L}_{\text{align}}$$

where $\lambda \in [0, 1]$ and $\lambda_{align} > 0$ are hyperparameters.

Fig. 2 visualizes the t-SNE plot of the learned embeddings. Although the figure and legends are dense, they shed valuable light on how the $\mathcal{E}$ space is organized. The plot is for 3 different floorplans and trajectories, but let's focus only on the bottom left corner, the region with circles. Two observations are of interest: (1) The green solid circle is a specific floorplan $\mathbf{x}$ and the orange and

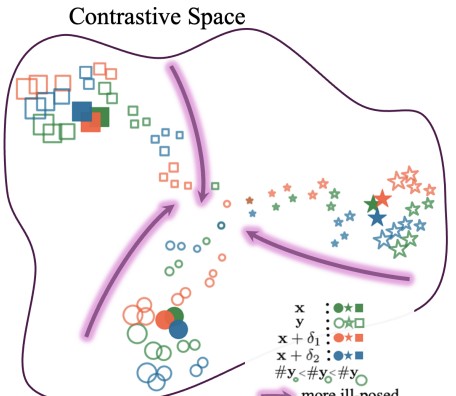

Figure 2: t-SNE embeddings from **CoGuide** for 3 floorplans (solid green) and two perturbed variants (solid orange/blue). Trajectories from these floorplans are shown as hollow shapes; larger hollow markers indicate higher trajectory density.

blue solid circles are slight variants of $\mathbf{x}$, denoted $\mathbf{x} + \delta_1$ and $\mathbf{x} + \delta_2$. Observe these variations are nearby while other floorplans (solid stars and solid squares) are far away. (2) The green circles are trajectories from the green floorplan (the same is true for other colors), and a larger radius indicates denser trajectories. Observe that trajectories are embedded near their matching floorplan, and sparser trajectories are further away from the floorplan (towards the center). This is expected because sparse trajectories imply more ill-posed behavior since many other floorplans can also explain those trajectories. We expect this organization to generate smoother likelihood scores, serving the original purpose of **CoGuide**.

---

**Algorithm 1 CoGuide**: Contrastive Diffusion Guidance for Spatial Inverse Problems

---

**Require:** $T$ timesteps; trajectory $\mathbf{y}$; step sizes $\{\zeta_t\}$; Adam base LR $\eta_0$, $\gamma_1,\gamma_2$, $\varepsilon$, noise scales $\{\tilde{\sigma}_t\}$;
  diffusion params $\{\alpha_t,\bar{\alpha}_t\}$; score $s_\theta$; encoders $f_\varphi,g_\psi$; temperature $\tau$, intersection weight $\lambda_{int}$.

**Require: Annealing & gating:** start $t_s$, end $t_e$, min-LR $\rho$, stop $t_{\text{stop}}$

1: $\mathbf{x}_T \sim \mathcal{N}(\mathbf{0},\mathbf{I})$;  Adam: $\boldsymbol{m} \leftarrow \mathbf{0}, \boldsymbol{v} \leftarrow \mathbf{0}$

2: **for** $t = T-1$ **down to** $0$ **do**

3: **DDIM:** $\hat{s} \leftarrow s_\theta(\mathbf{x}_t,t)$; $\hat{\mathbf{x}}_0 \leftarrow \bar{\alpha}_t^{-1/2}\big(\mathbf{x}_t + (1-\bar{\alpha}_t)\hat{s}\big)$;     ▷ score model and one step denoising

  $\hat{\epsilon} \leftarrow \dfrac{\mathbf{x}_t - \sqrt{\bar{\alpha}_t}\hat{\mathbf{x}}_0}{\sqrt{1-\bar{\alpha}_t}}$; $\mathbf{z} \sim \mathcal{N}(\mathbf{0},\mathbf{I})$. $\sigma_t \leftarrow \tilde{\sigma}_t$.

  $\mathbf{x}'_{t-1} \leftarrow \sqrt{\bar{\alpha}_{t-1}}\hat{\mathbf{x}}_0 + \sqrt{1-\bar{\alpha}_{t-1}-\sigma_t^2}\hat{\epsilon} + \sigma_t\mathbf{z}$.       ▷ ddim step

4: **CoGuide:** $G_t \leftarrow -\dfrac{1}{2\tau}\nabla_{\mathbf{x}_t}\|g_\psi(\mathbf{y}) - f_\varphi(\hat{\mathbf{x}}_0)\|_2^2 + \lambda_{int}\nabla_{\mathbf{x}_t}\|\mathbf{y}\odot(1-\hat{\mathbf{x}}_0)\|_1$.  ▷ contrastive likelihood score

5: **Adam:** $\eta_t \leftarrow \text{AnnealLR}(\eta_0,\rho,t;t_s,t_e)$; **if** $t_{\text{stop}}$ **set and** $t \geq t_{\text{stop}}$ **then** $\eta_t \leftarrow 0$.

  $\mathbf{x}_{t-1} \leftarrow \text{Adam}\big(\mathbf{x}'_{t-1},G_t; \eta_t,\gamma_1,\gamma_2,\varepsilon\big)$. *(Optional SGD: $\mathbf{x}_{t-1} \leftarrow \mathbf{x}'_{t-1} + \zeta_t G_t$.)*

6: **return** $\hat{\mathbf{x}}_0$

---

### 3.3 IMPROVING DIFFUSION INFERENCE

**Intersection Penalty.** We found it helpful to penalize intersections between *walls* and *trajectories* during inference. We add an intersection penalty $\mathcal{L}_{\text{intersect}} = \|\mathbf{y}\odot(1-\hat{\mathbf{x}}_0)\|_1$, to all baselines and **CoGuide**, which counts (up to a constant scale) the total number of pixels where a trajectory overlaps a wall. Minimizing this penalty term during reverse diffusion steers updates toward wall–trajectory compatibility, yielding floorplans that respect the observed trajectories $\mathbf{y}$.

**Using Adam with DDIM.** Once the likelihood surrogate and intersection penalty are plugged in, **CoGuide** performs gradient-based optimization over a nonconvex posterior via DDIM Song et al. (2020a) or DDPM Ho et al. (2020). Since DDIM uses fewer reverse steps than DDPM, plain GD/SGD can under-integrate our embedding-based gradients, leading to poor convergence. We therefore replace GD/SGD inside each DDIM step with *Adam* Kingma & Ba (2017). This supplements the reverse diffusion process with higher-order information about the optimization landscape and improves convergence. Algorithm 1 reflects this change by using Adam in the guidance step. To control guidance strength over the short DDIM schedule, we use a brief cosine annealing (denoted as AnnealLR in Algorithm 1) of the learning rate. *Before* the ramp starts (for $t \leq t_s$) we keep the rate fixed at $\eta_t = \eta_0$; *after* the ramp ends (for $t \geq t_e$) we clamp it to $\eta_{\min} = \rho\eta_0$. During the ramp ($t_s < t < t_e$) we use:

$$\eta_t = \eta_{\min} + \tfrac{1}{2}(\eta_0 - \eta_{\min})\Big[1 + \cos(\pi \tfrac{t-t_s}{t_e-t_s})\Big]. \tag{13}$$

Finally, we hard-gate guidance off by setting $\eta_t = 0$ for $t \geq t_{\text{stop}}$. This pairing, Adam for robust, per-coordinate integration and a short cosine ramp with a hard stop, recovers much of the "many-step" integration that DDPM would provide while preserving DDIM's speed. In addition, it avoids late-stage instabilities by letting the diffusion prior refine the sample without additional guidance.

**Using conditional diffusion priors.** (**CFG+CoGuide**) Using conditional diffusion priors, we further boost performance by combining a CFG-trained diffusion model $s_\theta(\mathbf{x}_t,t,\mathbf{c})$ with **CoGuide** so they can complement each other's strengths. In this setup, CFG acts as a conditional prior over floorplans given trajectory measurements $\mathbf{c}$; the only change is that we replace **CoGuide**'s original diffusion prior with the CFG model. When the null input $\mathbf{c} = \varnothing$, we recover the original **CoGuide** behavior.

## 4 EXPERIMENTS AND EVALUATION

**Blind Inverse Problems. CoGuide** extends naturally to blind inverse problems. We demonstrate this in Appendix D through restoration of historical audio recordings containing fully unknown degradations.

**Datasets.** We use the HouseExpo dataset Tingguang et al. (2019) for all experiments. It contains approximately 35,126 2D *floorplans* of houses and apartments generated from the SUNCG dataset Song et al. (2017). We downsample each floorplan to a $64 \times 64$ binary image where white pixels represent free space and black pixels represent walls/obstacles. Downsampling is necessary because path-planners scale poorly

to higher image sizes making diffusion inference slow. We split the dataset into an 80-10-10 for training, validation, and testing. Next, we generate compatible *trajectories* on each floorplan using the A* algorithm by randomly sampling start and goal locations from the open spaces. In addition, we generate trajectories at three levels of densities; sparse, moderate and dense to simulate how much a user may have walked. On average, these correspond to $40\%, 25\%$, and $10\%$ of the open spaces in the floorplans, respectively. Including this in the dataset helps in understanding how **CoGuide** copes with ill-posedness as the number of measured trajectories decreases.

**Metrics.** We evaluate the performance of different methods using 2 metrics: **(A) Intersection over Union (IoU)** Rezatofighi et al. (2019): This metric measures how well the predicted floorplan overlaps with the ground truth. It is computed as the ratio of intersecting free space pixels to the union of all free space pixels: $\text{IoU} = \frac{|\text{FP} \cap \text{FP}^*|}{|\text{FP} \cup \text{FP}^*|}$ where FP and FP$^*$ are the predicted and true free space pixels, respectively. **(B) F1 score** Sokolova & Lapalme (2009): Defined as $\text{F1} = \frac{2 \times P \times R}{P+R}$, where P is the *precision* and R is the *recall* of the bitmap. P and R are defined based on free space pixels, similar to IoU.

**Baselines.** We evaluate the performance of **CoGuide** against 6 competitive baselines: ■ **DPS+X**: DPS Chung et al. (2022) based inverse solver with 3 differentiable path-planners $\mathcal{A}$ by instantiating $\mathbf{X} \in$ {NeuralA*,TransPath,DiPPeR} Yonetani et al. (2021), Kirilenko et al. (2023), Liu et al. (2024). More details about the planners are provided in the Appendix E. Briefly, these path-planners use a CNN-based shortest-path module, a Transformer based path-probability encoder, and a diffusion-based planner, respectively. ■ **DiffPIR** Zhu et al. (2023): A plug-and-play image restoration solver that uses a diffusion model as a denoiser, instead of training a Gaussian denoiser. We follow the default configuration given in their publicly available codebase. Since we do not have a closed form data proximal estimator, we use an ADAM optimizer to solve the proximal with $10$ optimization steps and use a $0.01$ learning rate. ■ **DMPlug** Wang et al. (2024): A recent inverse solver that optimizes the noise seed such that after a DDIM sampler, the resulting image satisfies the measurement constraint. We follow the default configurations in their publicly available codebase except we use 100 optimization steps. ■ **CFG**: Performs Classifier-free Guidance Ho & Salimans (2022) that combines the unconditional and trajectory-conditioned scores with a guidance scale that biases sampling toward trajectory-consistent floorplans.

**Real-world Evaluation.** To evaluate real-world performance, we collected trajectories in two student apartments using the Qorvo DWM3000 ultrawideband (UWB) module Qorvo (2021). The UWB system consists of multiple anchors statically placed at known 3D coordinates in the apartment. The user carries a beacon that periodically pings each anchor, allowing us to estimate the time-of-flight (ToF) to that anchor. Using the ToF to line-of-sight anchors and the known speed of light, we apply trilateration to estimate the user's 3D position. We then project the 3D position onto the 2D floorplan to obtain walking trajectories. The UWB setup provides locations with a standard deviation of 10 cm. It is worth noting that the trajectories obtained via this method are sparser and distributed somewhat differently compared to our A*-generated dataset.

**Diffusion and Contrastive Model.** Our base diffusion model follows the implementation used in the DPS Chung et al. (2022). We adapted the contrastive model used in Khosla et al. (2020) based on the specific needs of this project. Classifier-free guidance (CFG) from the diffusers library is implemented by training a single UNet denoiser that can operate in both conditional and unconditional modes, and then combining its two outputs at inference with a guidance scale. The underlying idea and architecture come from the original classifier-free guidance paper Ho & Salimans (2022). We discuss architecture and hyperparameter settings for all our models in Appendix C.

## 4.1 RESULTS

■ **Quantitative.** Table 1 reports F1/IoU (mean $\pm$ std) for three regimes of trajectory density: sparse, moderate and dense. In the sparse regime, **CoGuide** attains the best performance, exceeding all baselines, including CFG. Similarly in the moderate regime, **CoGuide** again leads surpassing CFG and the DPS variants. However, in the dense case, CFG is the strongest, with **CoGuide** showing comparable performance. Overall, **CFG+CoGuide** consistently outperforms CFG, DPS+X, DiffPIR, and DMPlug.

■ **Qualitative.** Fig. 3 shows qualitative results of **CoGuide** and baselines on 6 test floorplans. The ground truth floorplan is shown in the top row with the measured trajectory **y** marked in light blue. The DPS+planner-based methods along with DiffPIR and DMPlug fail to produce valid floorplans that are consistent with the provided trajectory, often generating artifacts on the converged floorplan. This can be

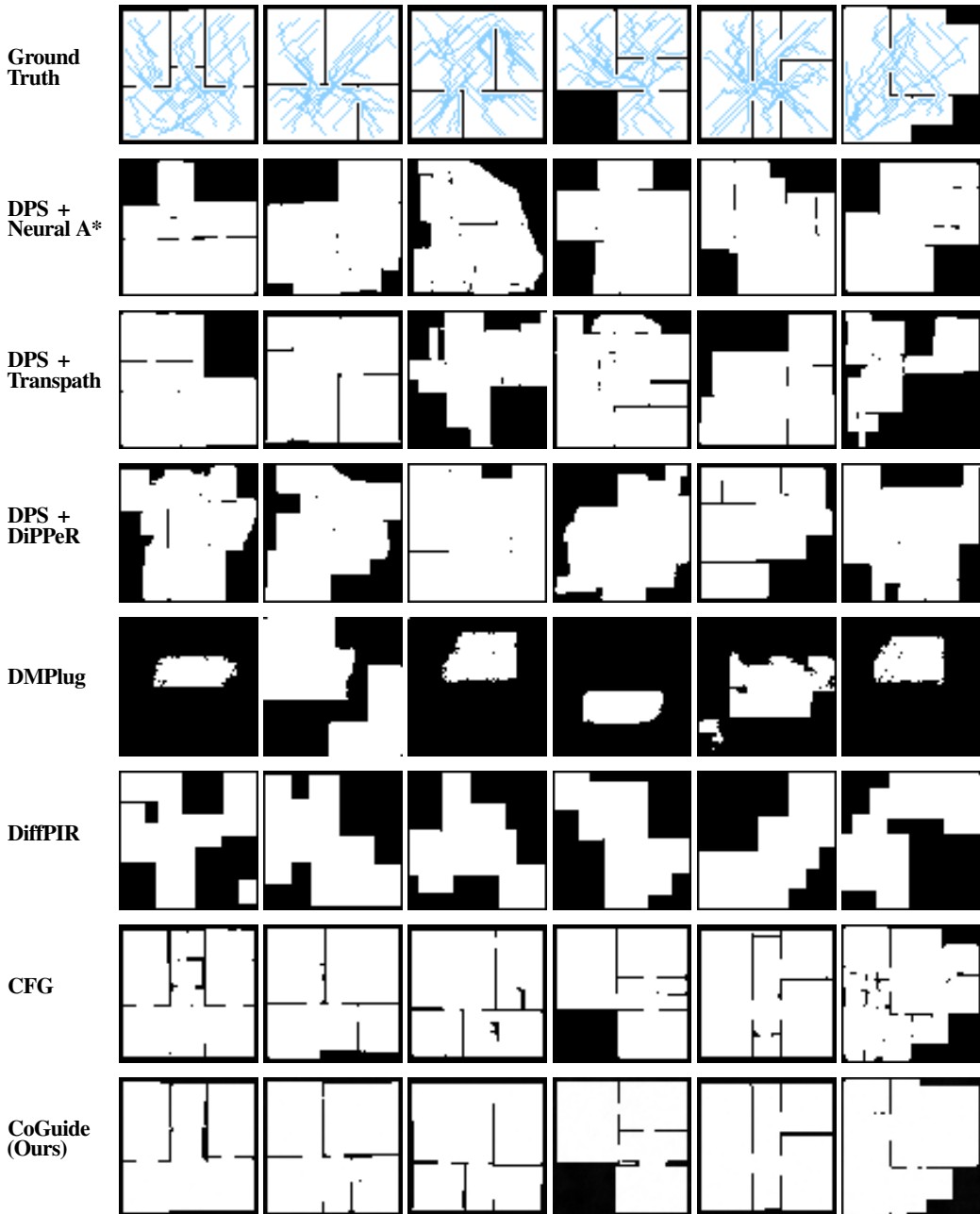

Figure 3: Qualitative comparison of ground truth floorplans against baselines and **CoGuide**.

attributed to the unstable forward operators $\mathcal{A}(.)$ that get embedded in each of their optimization processes. It is worth noting that although CFG proves superior in metrics, qualitative results do not always reflect this fact. **CoGuide** shows floorplans that are consistent with the trajectory, with fewer visual artifacts. We show additional results on different floorplans in Appendix B

## 4.2 ABLATIONS

■ **Effect of the Intersection Penalty.** As described in section 3.3, we add an intersection penalty to guide the posterior sampling to improve our results. Table 2 shows the effect of this intersection penalty on the results. We observe two points: (i) Inclusion of the intersection penalty $\mathcal{L}_{intersect}$ leads to an improvement in the evaluation metrics and the qualitative results. (ii) Applying too large of a penalty, however, degrades the results, as can be seen in the last row when $\lambda_{\text{int}} = 1.5 \times 10^{-3}$.

Table 1: Performance across trajectory densities. Each regime reports F1 and IoU (higher is better).

| Method | Sparse | | Moderate | | Dense | |
|---|---|---|---|---|---|---|
| | F1 | IoU | F1 | IoU | F1 | IoU |
| DPS+Neural A$^\star$ | 0.79±0.09 | 0.67±0.13 | 0.79±0.09 | 0.66±0.13 | 0.79±0.09 | 0.66±0.12 |
| DPS+TransPath | 0.76±0.15 | 0.64±0.18 | 0.74±0.15 | 0.60±0.19 | 0.72±0.17 | 0.59±0.20 |
| DPS+DiPPeR | 0.77±0.10 | 0.64±0.13 | 0.77±0.11 | 0.64±0.14 | 0.76±0.11 | 0.63±0.14 |
| DMPlug | 0.31±0.10 | 0.19±0.08 | 0.28±0.09 | 0.17±0.07 | 0.28±0.08 | 0.16±0.07 |
| DiffPIR | 0.63±0.09 | 0.47±0.09 | 0.64±0.08 | 0.48±0.09 | 0.65±0.08 | 0.49±0.08 |
| CFG | 0.86±0.06 | 0.76±0.10 | 0.93±0.03 | 0.88±0.05 | 0.97±0.01 | 0.95±0.03 |
| **CoGuide** | 0.91±0.04 | 0.84±0.07 | 0.94±0.03 | 0.89±0.05 | 0.95±0.03 | 0.90±0.06 |
| **CFG+CoGuide** | **0.93±0.04** | **0.87±0.07** | **0.97±0.03** | **0.93±0.04** | **0.99±0.01** | **0.97±0.02** |

■ **Improving convergence with Adam**
As discussed in section 3.3, we incorporate the Adam update during the sampling process and compare against the standard Gradient descent-based update. Results reported in Table 3 show that employing Adam consistently outperforms SGD across both DDPM and DDIM.

■ **Measurement Noise.** In real-world settings, the localization sensors that provide the input trajectories to **CoGuide** may be noisy. To model this, we inject Gaussian noise into the trajectory generation process. We increase the noise standard deviation and compare our performance across various noise levels and trajectory densities. **CoGuide**'s performance understandably degrades with increasing noise levels as shown in Fig. 4. The degradation is, however, graceful, and denser trajectory upholds better performance.

Table 2: **CoGuide** performance across intersection weight $\lambda_{int}$ settings.

| $\lambda_{int}$ | Sparse | | Moderate | | Dense | |
|---|---|---|---|---|---|---|
| | F1 | IoU | F1 | IoU | F1 | IoU |
| 0.0 | 0.88±0.05 | 0.78±0.08 | 0.90±0.04 | 0.82±0.07 | 0.91±0.04 | 0.84±0.07 |
| $3.0\times10^{-4}$ | 0.91±0.04 | 0.83±0.07 | 0.93±0.03 | 0.88±0.05 | 0.95±0.03 | 0.90±0.05 |
| $7.0\times10^{-4}$ | **0.91±0.04** | **0.84±0.07** | **0.94±0.03** | **0.89±0.05** | **0.95±0.03** | **0.90±0.06** |
| $1.5\times10^{-3}$ | 0.91±0.04 | 0.84±0.07 | 0.93±0.04 | 0.88±0.06 | 0.94±0.05 | 0.89±0.08 |

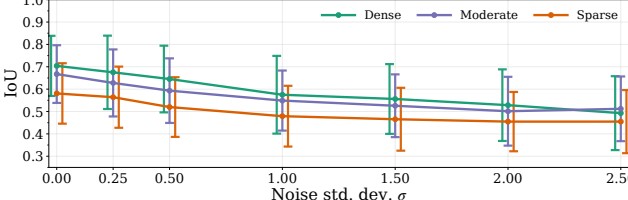

Figure 4: Effect of measurement noise on IoU.

Table 3: F1 and IoU for two optimizers under DDPM and DDIM samplers.

| Sampler | Method | F1 | IoU |
|---|---|---|---|
| DDPM | GD | 0.92±0.04 | 0.87±0.07 |
| | Adam | **0.94±0.03** | **0.88±0.06** |
| DDIM | GD | 0.86±0.07 | 0.76±0.10 |
| | Adam | **0.92±0.05** | **0.85±0.08** |

## 4.3 REAL-WORLD FLOORPLANS

We compare **CoGuide** and CFG on real-world floorplans using UWB measured trajectories. Although both methods make errors, **CoGuide** substantially infers more wall segments and room structure, whereas CFG often misses or misplaces major walls (Fig. 5). This gap is expected since CFG is explicitly trained to model the posterior density under the synthetic A*-generated trajectory distribution, and therefore struggles to generalize when faced with the sparser, differently distributed real-world trajectories.

*Additional inference results* on more floorplans and uncertainty analysis are provided in Appendix B. Finally, in Appendix D we also demonstrate *generalization* of **CoGuide** beyond spatial inverse problems by tackling a blind audio enhancement application. Audio demos are provided on our website.

## 5 RELATED WORK

**Floorplan estimation** has been studied extensively for a range of applications including room/graph reconstruction, layout parsing, and indoor mapping Lee et al. (2017); Gillsjö et al. (2023); Yang et al.

| Apartment | Human Walking | Trajectories | CFG | CoGuide |
|---|---|---|---|---|

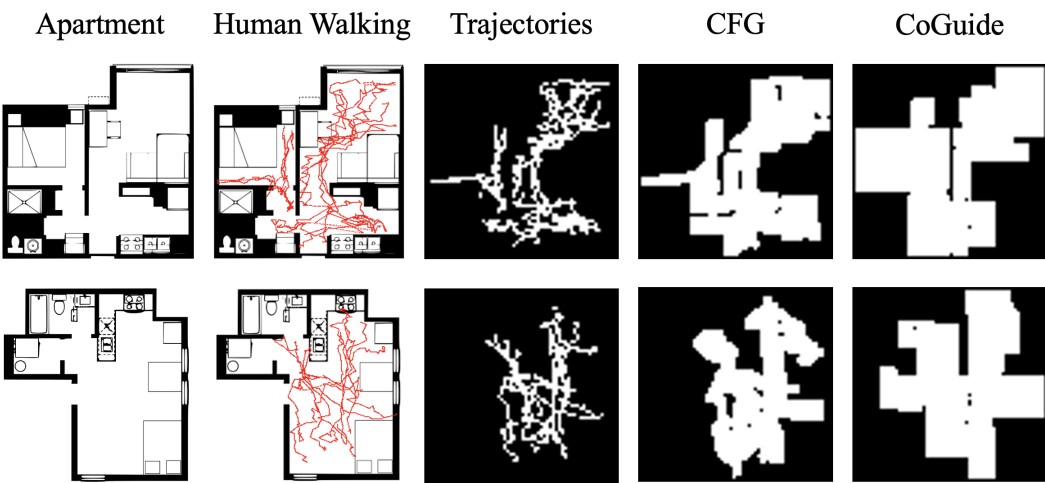

Figure 5: Real-world results comparing CFG and **CoGuide** on two apartment layouts. **CoGuide** reconstructed floorplans are qualitatively superior to those of CFG, which shows spurious structures.

(2023); Zou et al. (2018). Early works include classical and unsupervised pipelines leveraging mobile sensing and heuristics Shin et al. (2011). Graph-based methods have also been explored Yang et al. (2023); Hickman & Krolik (2009), modeling spatial relations via polygons, and wireframes.

**Vision-based** approaches using RGB images remain common and highly effective Lee et al. (2017); Zou et al. (2018); Zeng et al. (2019); Lv et al. (2021); Yan et al. (2020); Jia et al. (2022). A large body of work reconstructs floorplans from RGB images or panoramas via corner/edge decoding or Manhattan layouts Lee et al. (2017); Zou et al. (2018); Lv et al. (2021); Zeng et al. (2019), with extensions to 3D room layout from a single view Yan et al. (2020); Jia et al. (2022). More recently, diffusion models have emerged as strong priors for layout synthesis and reconstruction, including constrained or vectorized floorplan generation Gueze et al. (2023); Shabani et al. (2023); Inoue et al. (2023), leveraging advances in score-based modeling Ho et al. (2020); Song et al. (2020a); Rombach et al. (2022). While these visual pipelines are effective when imagery is available, they raise privacy concerns and depend on line-of-sight and scene illumination. In contrast, trajectories are privacy-preserving, and can be easily collected during routine motion using built-in IMU sensors on mobile devices.

**Other modalities** have also been explored beyond vision, including acoustics Zhou et al. (2017), magnetics Luo et al. (2017), RF Peng et al. (2018); Amballa et al. (2025), radar Hickman & Krolik (2009). While effective in specific settings (e.g., BatMapper Zhou et al. (2017)), many require specialized hardware or calibrated infrastructure, whereas IMU-based trajectories are easy to obtain. Walk2map Mura et al. (2021) is the most relevant to our work, but it is designed only for single-room floorplan layouts and does not have any underlying generative capabilities.

## 6 CONCLUSION AND FOLLOW-ON WORK

■ **Spatial Inverse Problems**: This paper focused on a specific floorplan estimation problem, however, the notion of contrastive guidance should lend itself to a broader family of non-differentiable $\mathcal{A}$ operators. We intend to investigate what family of operators can benefit, and conversely, how can the contrastive guidance be improved to broaden that family of operators. Along these lines, newer applications are also of interest. For instance, can city maps be synthesized based on GPS trajectories of vehicles? Can discrete molecular structure be synthesized from measured properties of molecules? Can Internet topologies be derived based on streaming packet analytics? ■ **Towards Realism**: Even in our specific floorplan application, there is room for improvement. The standard floorplan dataset (HouseExpo) does not include furniture. Incorporating furniture into the environment is a practical extension, if such a dataset is available. ■ **Blind Inverse Problems**: This paper showed early results on solving blind inverse problems in the audio domain. We believe this offers promise that contrastive guidance can enable a line of attack for complicated blind inverse problems, as long as we can generate measurement $\mathbf{y}$ from known $\mathbf{x}$'s. If partial information is available about $\mathcal{A}(.)$, could that be adequate to design a good embedding space? We believe **CoGuide** could initiate conversation along all these branches, inviting a range of follow-on research and exploration.

ACKNOWLEDGMENTS

We are grateful to Sahil Bhandary Karnoor for helping us collect real-world data. We thank the anonymous reviewers for their valuable feedback. This work was partially supported by NSF #2008338, #1909568, #2148583, and #MRI-2018966. This work used DELTA at NCSA through allocation CIS230230 from the Advanced Cyberinfrastructure Coordination Ecosystem: Services & Support (ACCESS) program, which is supported by U.S. National Science Foundation grants #2138259, #2138286, #2138307, #2137603, and #2138296

## 7 REPRODUCIBILITY STATEMENT

All model implementations, training scripts, and inference pipelines are open-sourced and will be made available in our anonymous GitHub repository. This includes instructions for environment setup, dependencies, and reproducible random seeds. The datasets used in our experiments are publicly available. We provide detailed descriptions of the dataset in Sec. 4. Hyperparameters, training schedules, and evaluation pipelines are described in Sec. 4, with further details in the Appendix C. Additional information including code repository can be found at `https://coguide.github.io/`.

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

APPENDIX

## A  LIKELIHOOD UNDER INFONCE OBJECTIVE

**CoGuide** benefits from the replacement of the likelihood term with a contrastive similarity score from the InfoNCE formulation. This relationship was originally shown in Contrastive Predictive Coding Oord et al. (2018). We include this derivation here for completeness in the context of **CoGuide**. Consider a batch of size $N$ with floorplans $\widetilde{\mathcal{X}} = \{\mathbf{x}_j\}_{j=1}^N$ and trajectories $\widetilde{\mathcal{Y}} = \{\mathbf{y}_j\}_{j=1}^N$. For a given floorplan $\mathbf{x}_j$, we shuffle the trajectories $\widetilde{\mathcal{Y}}$, so that the corresponding positive trajectory $\mathbf{y}_j$ is placed at position $i \in \{1,...,N\}$. The task is then to identify the correct index $i$. Let $I$ be the random variable representing the index of the correct trajectory for floorplan $\mathbf{x}_j$ and $p(\mathbf{y})$ be the marginal distribution of trajectories. Given a similarity general score function $s(\mathbf{x},\mathbf{y})$, the InfoNCE assumes a softmax distribution over indices and formulates a cross-entropy loss. So, the approximate posterior $q$:

$$q(I = i \,|\, \mathbf{x}_j, \widetilde{\mathcal{Y}}) = \frac{\exp\{s(\mathbf{x}_j,\mathbf{y}_i)\}}{\sum_{k=1}^N \exp\{s(\mathbf{x}_j,\mathbf{y}_k)\}}. \tag{14}$$

Specifically, in **CoGuide**, the similarity score function assumes the form $s(\mathbf{x},\mathbf{y}) = \langle f_\varphi(\mathbf{x}), g_\psi(\mathbf{y}) \rangle / \tau$ since we use encoders $f_\varphi$ (for floorplans), and $g_\psi$ (for trajectories) as described in section 3.2.

Next, we calculate the true optimal posterior $p$ from Bayes' rule as :

$$p(I=i\,|\,\mathbf{x}_j,\widetilde{\mathcal{Y}}) = \frac{p(\mathbf{x}_j,\widetilde{\mathcal{Y}},I=i)}{\sum_{r=1}^N p(\mathbf{x}_j,\widetilde{\mathcal{Y}},I=r)}$$

$$= \frac{\frac{1}{N}p(\mathbf{y}_i\,|\,\mathbf{x}_j)\prod_{k\neq i}p(\mathbf{y}_k)}{\sum_{r=1}^N \frac{1}{N}p(\mathbf{y}_r\,|\,x_j)\prod_{k\neq r}p(\mathbf{y}_k)}$$

$$= \frac{p(\mathbf{y}_i\,|\,\mathbf{x}_j)\prod_{k\neq i}p(\mathbf{y}_k)}{\sum_{r=1}^N p(\mathbf{y}_r\,|\,\mathbf{x}_j)\prod_{k\neq r}p(\mathbf{y}_k)}$$

$$= \frac{\dfrac{p(\mathbf{y}_i\,|\,\mathbf{x}_j)}{p(\mathbf{y}_i)}}{\sum_{r=1}^N \dfrac{p(\mathbf{y}_r\,|\,\mathbf{x}_j)}{p(\mathbf{y}_r)}}.$$

Matching $q$ and $p$ requires that the numerators differ only by a multiplicative constant (since softmax is invariant to shifts). Thus, at the InfoNCE optimum,

$$\exp\big(\langle f_\varphi(\mathbf{x}),g_\psi(\mathbf{y})\rangle/\tau\big)\propto\frac{p(\mathbf{y}\,|\,\mathbf{x})}{p(\mathbf{y})}$$

$$\implies \frac{1}{\tau}\langle f_\varphi(\mathbf{x}),g_\psi(\mathbf{y})\rangle\propto\log p(\mathbf{y}\,|\,\mathbf{x})-\log p(\mathbf{y})$$

$$\implies \frac{1}{\tau}\langle f_\varphi(\mathbf{x}),g_\psi(\mathbf{y})\rangle=\log p(\mathbf{y}\,|\,\mathbf{x})-\log p(\mathbf{y})+C \tag{15}$$

where $C$ is independent of $\mathbf{x}$. Taking gradients with respect to $\mathbf{x}$ cancels both $\log p(\mathbf{y})$ and $C$, yielding

$$\frac{1}{\tau}\nabla_x\langle f_\varphi(\mathbf{x}),g_\psi(\mathbf{y})\rangle = \nabla_\mathbf{x}\log p(\mathbf{y}\,|\,\mathbf{x}). \tag{16}$$

Therefore, we substitute this surrogate likelihood into Eq. 5) at time-$t$ along with the DPS approximation which gives,

$$\nabla_{\mathbf{x}_t}\log p_t(\mathbf{x}_t|\mathbf{y})\approx s_\theta(\mathbf{x}_t,t)+\frac{1}{\tau}\nabla_{\mathbf{x}_t}\langle f_\varphi(\hat{\mathbf{x}}_0(\mathbf{x}_t)),g_\psi(\mathbf{y})\rangle$$

$$= s_\theta(\mathbf{x}_t,t)+\frac{1}{\tau}\nabla_{\mathbf{x}_t}\big(1-\frac{1}{2}\big\|f_\varphi(\hat{\mathbf{x}}_0(\mathbf{x}_t))-g_\psi(\mathbf{y})\big\|_2^2\big).$$

$$= s_\theta(\mathbf{x}_t,t)-\frac{1}{2\tau}\nabla_{\mathbf{x}_t}\big\|f_\varphi(\hat{\mathbf{x}}_0(\mathbf{x}_t))-g_\psi(\mathbf{y})\big\|_2^2. \tag{17}$$

We replace the inner product with the $L_2$ norm since $\|f_\varphi(\mathbf{x})\|_2 = \|g_\psi(\mathbf{y})\|_2 = 1$. This derivation justifies that the InfoNCE objective ties together contrastive similarity with the true likelihood $p(\mathbf{y}|\mathbf{x})$. On extending this approach to supervised contrastive training, the validity of the likelihood surrogate $\nabla_{\mathbf{x}_t}\|f_\varphi(\hat{\mathbf{x}}_0(\mathbf{x}_t))-g_\psi(\mathbf{y})\|_2^2$ remains intact. This holds because, during diffusion inference, only a single measurement of $\mathbf{y}$ is provided to the model. Thus, the contrastive objective in this case uses only a single-positive sample, and therefore, reduces to that of the InfoNCE objective.

## B  ADDITIONAL QUALITATIVE RESULTS

■ **Additional Results.** We show results from evaluation on additional floorplans in Fig. 6 and 7. We observe that **CoGuide** significantly outperforms all inverse-solver baselines while being better or comparable to CFG.

■ **Uncertainty quantification.** We eventually imagine a user-in-the-loop system because not all floorplans are observed with the same number of trajectories; our estimates can vary and should reflect this effect. We therefore quantify an *uncertainty* in the predicted floorplan by drawing multiple posterior samples and computing the variance of its distance transform Strutz (2023) while allowing for small translations.

Fig. 9 shows the uncertainty in the predictions of **CoGuide** across varying trajectory densities for two different floorplans. As is expected, increasing trajectories lead to a reduction in the uncertainty, as

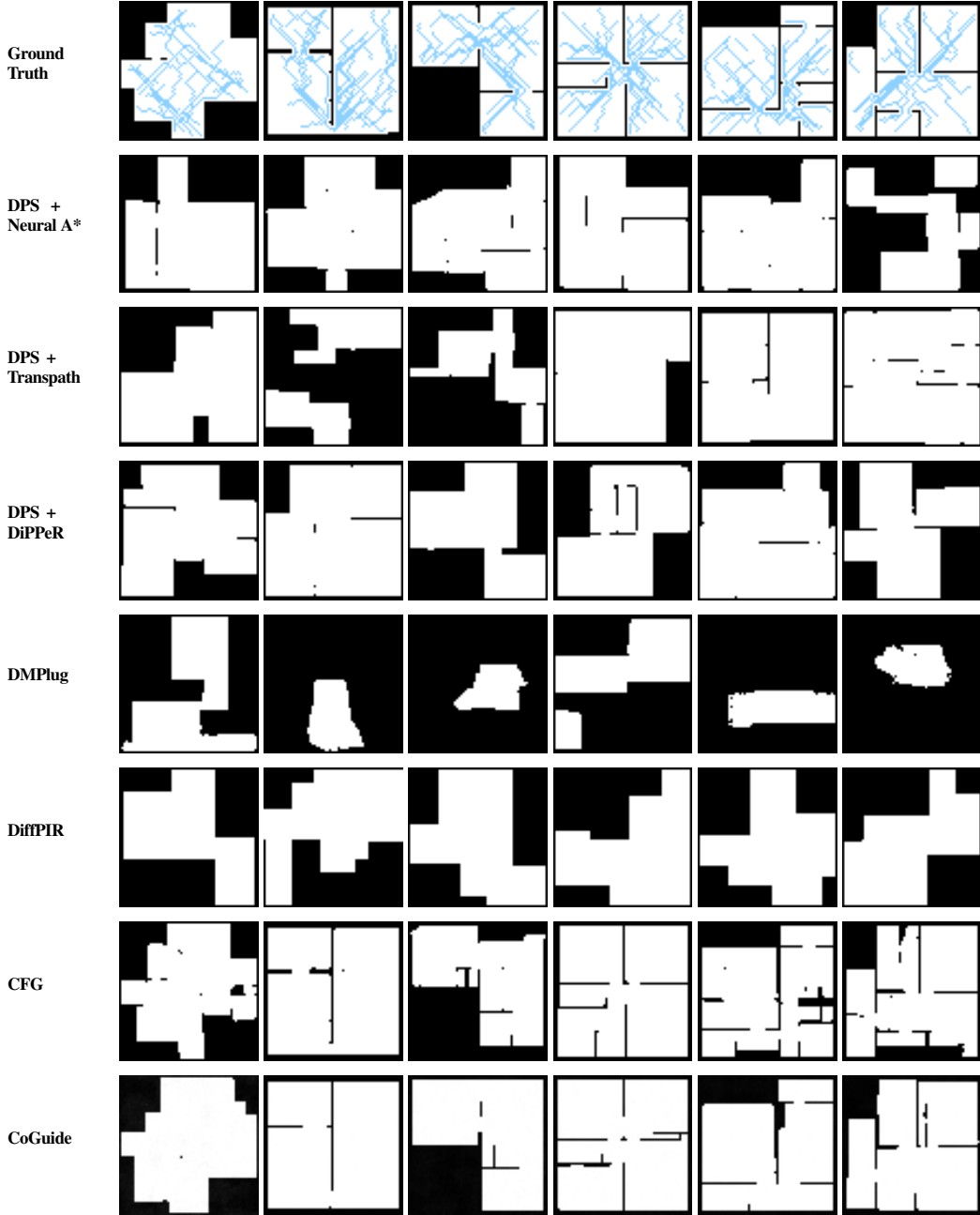

Figure 6: Qualitative comparison of ground truth floorplans against baselines.

highlighted by a low amount of red towards the bottom right of the images. In regions of high uncertainty (more red), a user in a practical-setting may collect more trajectory measurements in those regions. As the measurements increase, the uncertainty in the predictions decreases eventually converging to the true floorplan.

■ **Top 5-nearest floorplans in the contrastive embedding space** We include figures to show the closest 5 floorplans to the ground truth in the contrastive embedding space. It is evident that the contrastive space has learned to place semantically similar looking floorplans close-by. This further supports the original goal of **CoGuide**. We observe that as sparsity increases, the top-5 retrieved floorplans move away from the ground truth. This reflects the nature of the contrastive space that was shown in the t-SNE plot in Fig. 2

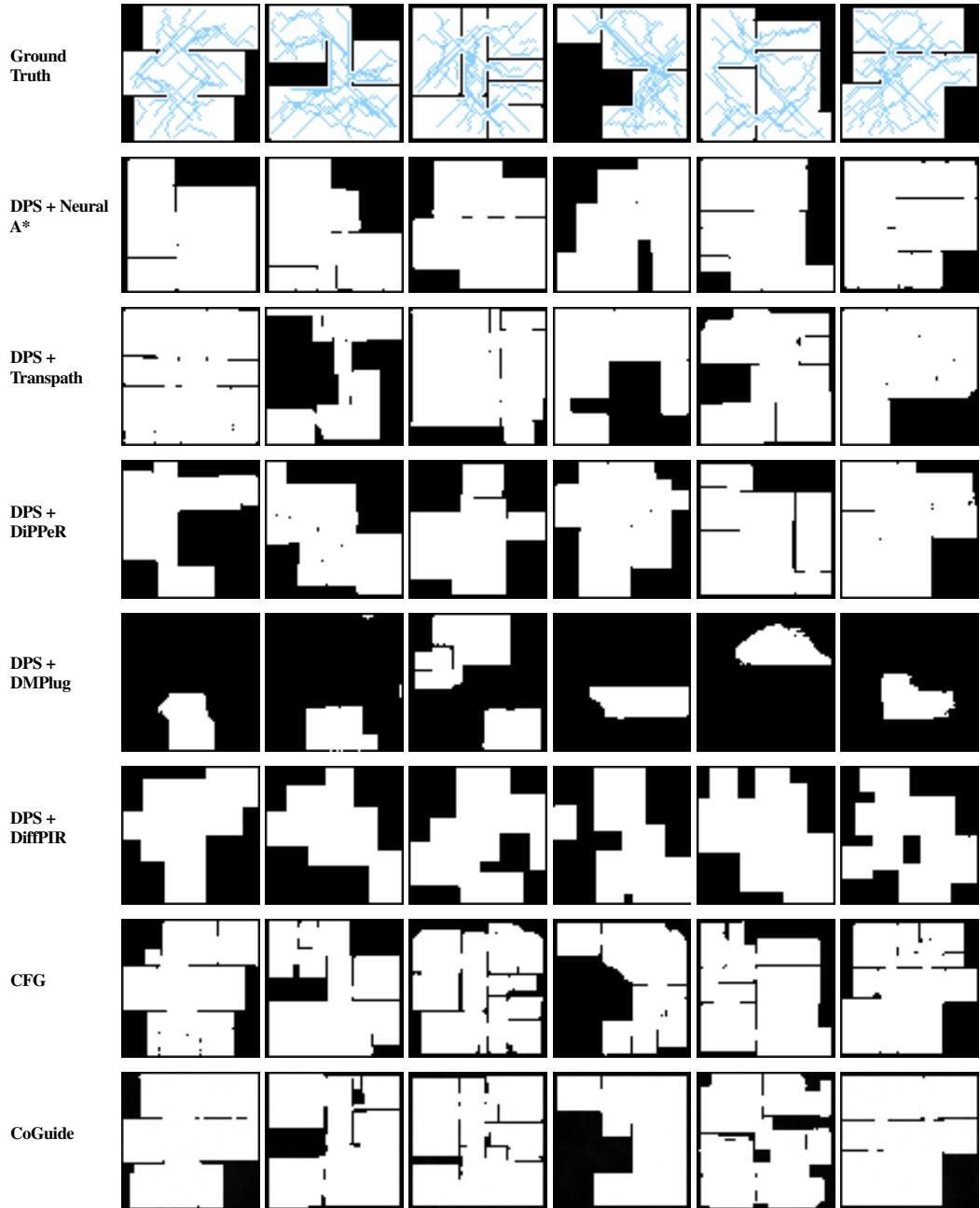

Figure 7: Additional qualitative comparison of ground truth floorplans against baselines.

## C  COGUIDE IMPLEMENTATION DETAILS

### C.1  DIFFUSION MODEL

**Inputs and outputs.** All models operate on single-channel floorplan images of spatial size $64 \times 64$. The network predicts a single-channel output and is trained to estimate the diffusion noise (i.e., $\epsilon$-parameterization). Training was done on NVIDIA RTX3090 GPU with 24GB RAM for around 28 hours.

**Backbone topology.** We use a U-shaped encoder–decoder with skip connections at every resolution. The base feature width at the first stage is 128 channels. The network has four resolution stages with channel

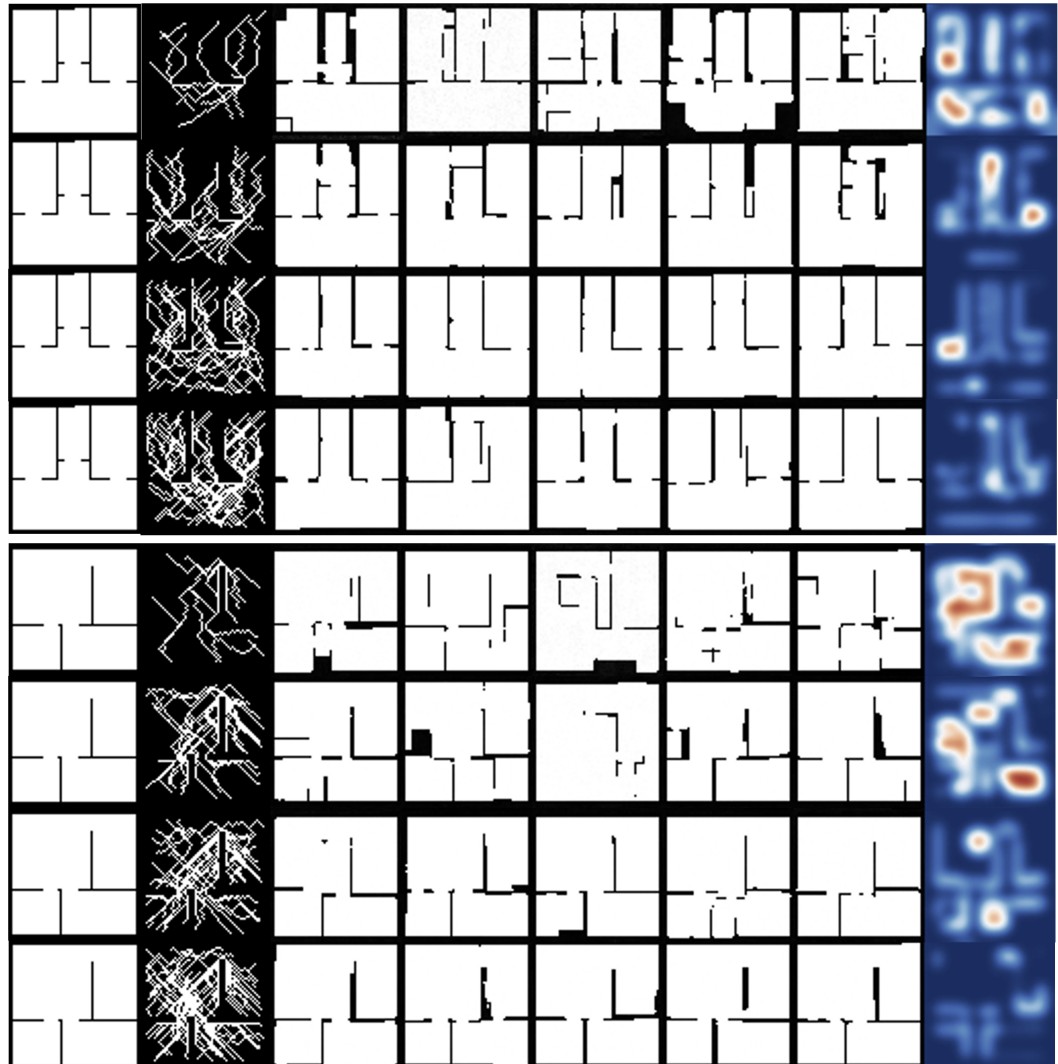

Figure 8: Variance of Distance Transform across 5 random seeds to quantify uncertainty. Decreasing uncertainty with increasing trajectory density marked is evident from the reduction in red regions.

multipliers $(1,2,3,4)$ applied to the base width as spatial resolution decreases, yielding encoder widths $(128,256,384,512)$ and a symmetric decoder. Each resolution stage contains one residual block per scale. Residual blocks follow the sequence: normalization layer, SiLU nonlinearity, $3\times3$ convolution, followed by a second normalization–SiLU–$3\times3$ stack inside the block; a skip projection is included when input and output widths differ. Spatial downsampling and upsampling are performed inside residual blocks (residual down/up blocks), keeping the skip topology consistent across scales.

**Timestep conditioning.** Diffusion timesteps are embedded with sinusoidal features and passed through a two-layer multilayer perceptron with SiLU activation. The embedding dimensionality equals four times the base feature width $(4\times128)$. This vector conditions every residual block through feature-wise affine modulation (scale and shift applied after normalization). When scale–shift modulation is disabled, the embedding is added to the block features instead.

**Self-attention.** Self-attention is inserted at the stage whose spatial size equals $64/16\!=\!4$ (i.e., after fourfold downsampling). Attention uses four heads with 64 channels per head. Query, key, and value projections are computed with $1\times1$ convolutions over the flattened spatial axis; the dot-product weights are scaled by

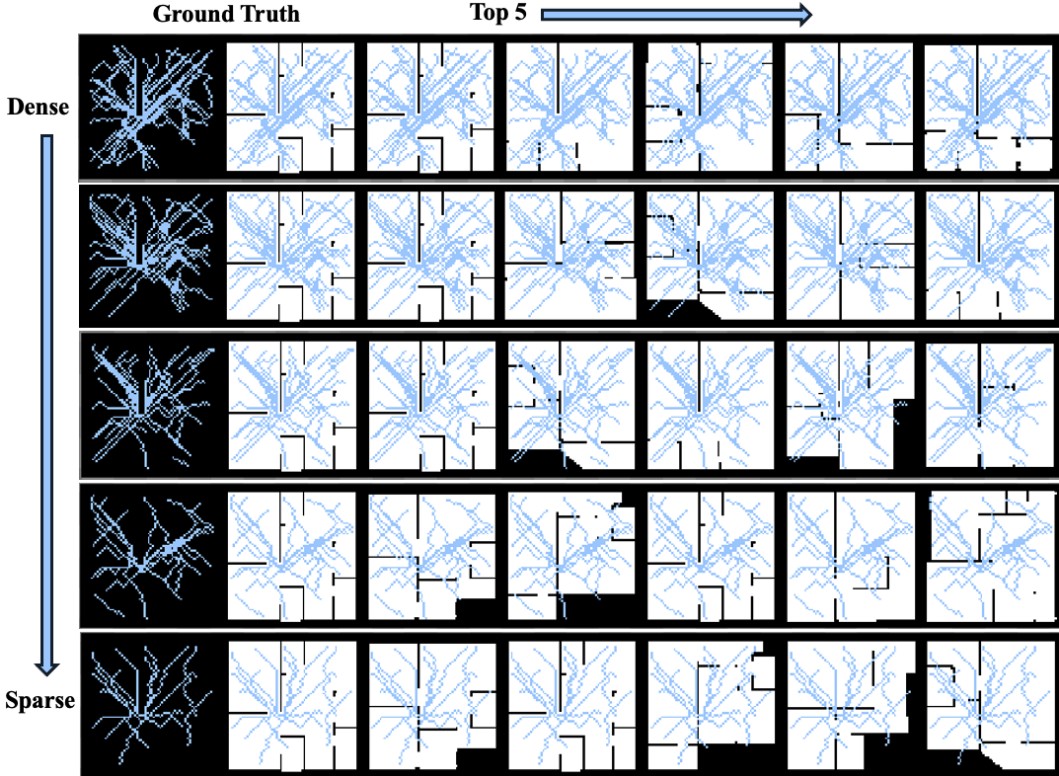

Figure 9: Retrieving floorplans corresponding to the top 5 nearest embeddings to the ground truth floorplan. The top-1 is always the ground truth itself.

$1/\sqrt{d}$ for stability, and the attended features are projected back to the model width with a $1\times1$ convolution. The same attention configuration is mirrored on the corresponding decoder stage.

**Normalization, activation, and regularization.** All blocks use the same normalization layer before SiLU activations. Dropout is disabled (rate $=0$) throughout the network. Zero-initialized $3\times3$ convolutions are used at the end of residual blocks to stabilize early training.

**Output head.** After the final decoder block, a normalization and SiLU are applied, followed by a $3\times3$ convolution that maps the current feature width back to one channel. No auxiliary heads are used.

**Encoder-only and super-resolution variants.** For representation studies, we use an encoder-only variant that shares the same residual and attention layout, terminating in either adaptive average pooling or attention pooling to produce a compact vector embedding. For conditional super-resolution ablations, a second (low-resolution) image is bilinearly upsampled to the target size and concatenated with the noisy input along the channel dimension before entering the first convolution; the rest of the architecture is unchanged.

**Optimization and schedule.** Unless otherwise noted, we train for a long horizon (on the order of $10^3$ epochs) using AdamW with weight decay 0.05, an initial learning rate of $10^{-4}$, gradient–norm clipping at 1.0, and cosine decay to a zero floor following a short warmup of 1–2% of total steps. Exponential moving averaging of weights (decay 0.999–0.9999) can be enabled for evaluation stability; results are reported with and without EMA when relevant. Batch size is chosen to saturate device memory; when necessary, gradient accumulation is used to reach an effective batch size comparable across setups. Dropout within residual blocks is disabled (rate 0).

**Diffusion hyperparameters.** We use a linear variance schedule with $T=1000$ steps. Training targets are the additive noise (epsilon–parameterization). The predicted mean follows the standard epsilon formulation, and the variance is handled inside the diffusion objective with a learned–range parameterization; no auxiliary output heads are added. Denoised clipping is enabled, dynamic thresholding is disabled, and

timestep rescaling is not used. For sampling, we report both ancestral sampling using the full 1000–step trajectory and deterministic sampling using a 100–step respaced trajectory.

**Tunable knobs.** Architectural knobs include: base width (default 128), the four–stage multiplier tuple (1,2,3,4), number of residual blocks per stage (default one), attention placement (default only at $4 \times 4$), number of attention heads (default four), and head width (default 64). Training knobs include: total epochs (order of $10^3$), optimizer (AdamW), weight decay ($10^{-2}$–$10^{-1}$; default 0.05), initial learning rate ($5 \times 10^{-5}$ to $3 \times 10^{-4}$; default $10^{-4}$), warmup ratio (1–2%), cosine decay floor (zero), gradient–norm clip (default 1.0), EMA decay (0.999–0.9999), and batch size or effective batch size via accumulation. Diffusion–process knobs include: number of training steps $T$ (default 1000), schedule type (linear by default), respacing for sampling (full vs. 100–step DDIM).

## C.2   CLASSIFIER-FREE GUIDANCE

**Architecture.** Our diffusion prior uses a U-Net backbone with four resolution stages. The network operates on $64 \times 64$ inputs and increases its channel capacity across levels $(128, 256, 384, 512)$. Each resolution stage contains a single residual block, and a standard mid-block is placed between the downsampling and upsampling paths. The upsampling hierarchy mirrors the downsampling path to preserve multi-scale information. We use group normalization with 32 groups throughout the network, and time embeddings are injected via scale–shift modulation in each residual block so that the denoising behavior is explicitly conditioned on the diffusion timestep.

**Training.** We train the model for up to 50 epochs on $64 \times 64$ images with a batch size of 128. The learning rate is set to $10^{-4}$ with 500 warmup steps and no gradient accumulation. Classifier-free guidance Ho & Salimans (2022) is enabled by randomly dropping the conditioning with probability 0.1 during training, so the network learns both conditional and unconditional denoising behavior in a single model. Training is stopped at 50 epochs because the validation curve plateaus.

**Inference and guidance tuning.** At inference time, we run a standard diffusion sampler starting from Gaussian noise and iteratively denoising toward a clean floorplan. Given unconditional and conditional denoiser outputs, $s_\theta(\mathbf{x}_t, t, \varnothing)$ and $s_\theta(\mathbf{x}_t, t, \mathbf{c})$, we form the classifier-free guided prediction

$$s_{\theta,\mathrm{CFG}}(\mathbf{x}_t, t, \mathbf{c}) = (1+\gamma) s_\theta(\mathbf{x}_t, t, \mathbf{c}) - \gamma s_\theta(\mathbf{x}_t, t, \varnothing), \tag{18}$$

where $\gamma$ is the guidance scale. We tune $\gamma$ by sweeping over several values and selecting the setting $\gamma = 1.5$ that achieves the best validation performance (Fig. 10).

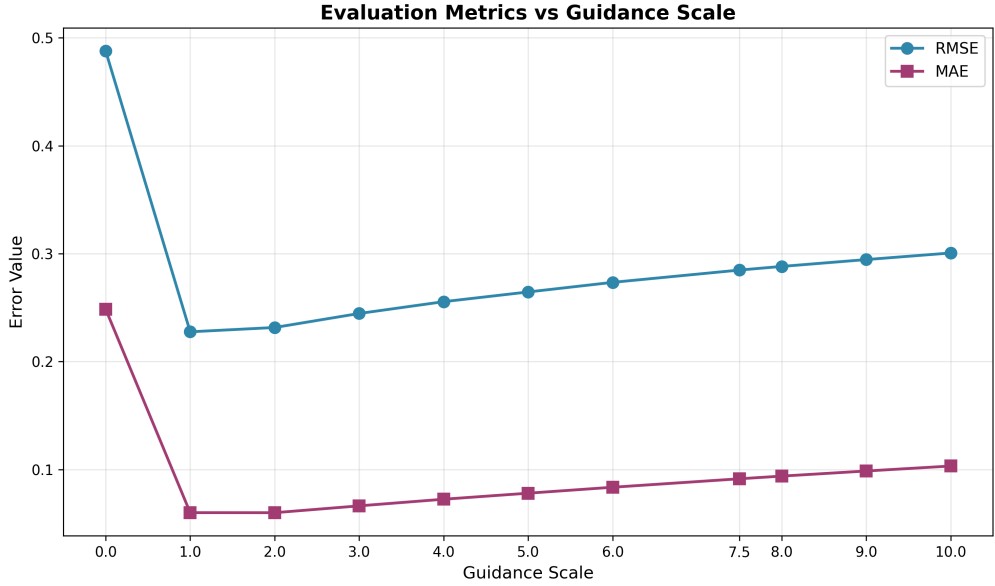

Figure 10: Tuning the guidance for inference in CFG

### C.3 Supervised Contrastive Learning

**Encoders and embeddings.** Both modalities (floorplan and trajectory) are processed by identical vision transformers operating at $64 \times 64$ resolution with $16 \times 16$ patches. The patch stem is adapted to one input channel. The transformer produces a global token which is mapped to a 256-dimensional space by a lightweight two-layer projection: linear $\rightarrow$ GELU $\rightarrow$ linear within 256 dimensions, followed by dropout ($p = 0.1$), a residual connection from the first linear output, and LayerNorm. The resulting vectors are $\ell_2$-normalized prior to similarity computation.

**Batch organization and positives.** Mini-batches of size 496 are constructed so that each floorplan anchor is paired with 7 trajectory positives from the same scene, yielding a multi-positive setting. All other examples in the batch that do not share the label act as negatives. Similarities are inner products between unit vectors.

**Data transformations.** To encourage invariance, random rotations and horizontal/vertical flips are applied to both modalities. For trajectories, structured dropout removes a uniformly sampled fraction of path pixels in the range $[0.05, 0.10]$ per sample. Augmentations are applied independently across views.

**Contrastive objective and temperature.** The supervised contrastive term uses a temperature-scaled log-softmax over all candidates in the batch. The temperature is learned during training, softly constrained to the interval $[0.01, 0.15]$, and updated with a small learning rate ($10^{-4}$). Unless stated, the contrastive weight is initially one and may be decayed later (see scheduling).

**Cross-modal alignment and directionality.** Alongside the symmetric multi-positive objective, we add a distance-based alignment penalty between matched floorplan–trajectory embeddings to co-locate corresponding pairs. A directional coupling that emphasizes trajectory-to-floorplan consistency is included with half the weight of the main contrastive term. A uniformity regularizer (energy-based) is available but disabled by default.

**Loss scheduling.** After an initial phase of 100 epochs, the contrastive weight is optionally reduced linearly over 30 epochs to a nonzero tail value (e.g., $0.5$), then held fixed. This schedule prioritizes discrimination early and refinement later. Alternative schedules that delay the alignment term use the same ramp length and maintain a reduced contrastive tail; the discrimination-first schedule is the default.

**Optimization and training hyperparameters.** Training runs for 1500 epochs with AdamW, weight decay $0.05$, and an initial learning rate of $10^{-4}$. Gradients are clipped to a global norm of $1.0$. The learning rate uses a short warmup covering $1.5\%$ of total steps, followed by cosine decay to a zero floor. The temperature parameter is optimized jointly under the same schedule.

**Optional stochastic floorplan corruption.** For ablations, we considered variance-preserving diffusion-style corruption applied to floorplan inputs with a small probability and limited diffusion time within a 1000-step schedule. This corruption is disabled in the main experiments to isolate the supervised contrastive contribution.

**Summary.** Single-channel inputs for both modalities; shared transformer encoders at $64 \times 64$ with 16-pixel patches; a residual two-layer projection to 256 dimensions with unit-length normalization; multi-positive supervised contrastive learning with a learned temperature; rotation/flip invariances and mild trajectory dropout; auxiliary alignment and directional terms with modest weights; and a long-horizon AdamW optimization with warmup and cosine decay. Training was done on NVIDIA A6000 GPUs with 48GB RAM for around 12 hours.

### C.4 Loss function descriptions

Here, we explicitly state the expressions for the contrastive losses that were used in the implementation of **CoGuide**. The losses include a symmetric supervised contrastive loss along with a positives-only alignment loss. Let $f_\varphi(\mathbf{x})$ be the floorplan encoder and $g_\psi(\mathbf{y})$ be the trajectory encoder. For each floorplan $\mathbf{x}_i$, there are $K$ matched trajectories $\{\mathbf{y}_{i,k}\}_{k=1}^{K}$. Negatives come from other items in the batch.

**Floorplan $\rightarrow$ Trajectory, $\mathcal{L}_{f \rightarrow t}$.** Anchor: the floorplan embedding $g_\psi(\mathbf{x}_i)$. Positives: its $K$ matched trajectory embeddings $f_\varphi(\mathbf{y}_{i,k})$. The numerator scores a true pair; the denominator sums scores over all

trajectories in the batch. This pulls in the right trajectories and pushes away the rest.

$$\mathcal{L}^{f \to t} = \frac{1}{B} \sum_{i=1}^{B} \left[ -\frac{1}{K} \sum_{k=1}^{K} \log \frac{\exp\left(\langle f_\varphi(\mathbf{x}_{i,k}), g_\psi(\mathbf{y}_i) \rangle\right)}{\sum_{j=1}^{B} \sum_{k'=1}^{K} \exp\left(\langle f_\varphi(\mathbf{x}_{i,k}), g_\psi(\mathbf{y}_i) \rangle\right)} \right].$$

**Trajectory $\to$ Floorplan, $\mathcal{L}_{t \to f}$.** Anchor: the trajectory embedding $f_\varphi(\mathbf{y}_{i,k})$. The numerator scores its matched floorplan $g_\psi(\mathbf{x}_i)$; the denominator sums over all floorplans in the batch. This pulls in the right floorplan and pushes away others. Using both directions keeps both encoders balanced.

$$\mathcal{L}^{t \to f} = \frac{1}{BK} \sum_{i=1}^{B} \sum_{k=1}^{K} \left[ -\log \frac{\exp\left(\langle f_\varphi(\mathbf{x}_{i,k}), g_\psi(\mathbf{y}_i) \rangle\right)}{\sum_{j=1}^{B} \exp\left(\langle f_\varphi(\mathbf{x}_{i,k}), g_\psi(\mathbf{y}_j) \rangle\right)} \right].$$

**Alignment loss, $\mathcal{L}_{\mathbf{align}}(\alpha)$.** Directly shrinks the distance of each matched pair $\|g_\psi(\mathbf{x}_i) - f_\varphi(\mathbf{y}_{i,k})\|_2^\alpha$ (often $\alpha = 2$). With $\ell_2$-normalized embeddings, this equals increasing cosine similarity.

$$\mathcal{L}_{\text{align}}(\alpha) = \frac{1}{BK} \sum_{i=1}^{B} \sum_{k=1}^{K} \left\| g_\psi(\mathbf{y}_i) - f_\varphi(\mathbf{x}_{i,k}) \right\|_2^2,$$

**Total loss, $\mathcal{L}_{\mathbf{total}}$.** We blend the two contrastive loss terms and the alignment term. In practice, we start with $\lambda_{\text{align}} = 0$ and increase it after a few epochs so the contrastive losses first separate negatives, then alignment tightens each true pair.

$$\mathcal{L}_{\text{contra}} = \lambda \mathcal{L}_{f \to t} + (1 - \lambda) \mathcal{L}_{t \to f} + \lambda_{\text{align}} \mathcal{L}_{\text{align}}.$$

Here, $\lambda \in [0,1]$ and $\lambda_{align} > 0$ are tunable hyperparameters.

# D GENERALIZING COGUIDE TO BLIND INVERSE PROBLEMS

**CoGuide**'s contrastive diffusion-guidance enables us to solve blind inverse problems. We apply it to a *blind audio restoration* task on historical piano recordings that are degraded by unknown operators $\mathcal{A}(\cdot)$ (including spectral coloration, additive noise and clicks, nonlinear effects, etc.). This setting is fully *blind* because we neither assume an analytical form for $A(\cdot)$ nor estimate it at inference time.

## D.1 FORMULATION.

Like in **CoGuide**, we formulate this problem with a forward model. Here an unknown clean audio waveform $\mathbf{x}$ is observed through an unknown degradation operator $\mathcal{A}$ plus noise $\mathbf{w} \sim \mathcal{N}(\mathbf{0}, \sigma^2 \mathbf{I})$:

$$\mathbf{y} = \mathcal{A}(\mathbf{x}) + \mathbf{w}. \tag{19}$$

Neither $\mathbf{x}$ nor $\mathcal{A}$ is known at test time. The goal is to recover a plausible $\hat{\mathbf{x}}$ that both explains the observation $\mathbf{y}$ and lies on the manifold of realistic audio samples. Following Karras et al. (2022) and the formulation in Moliner et al. (2024), we parameterize time by a continuous variable $\tau$, with $\mathbf{x}_\tau$ denoting the audio sample noisified at time $\tau$. At $\tau = T$, $\mathbf{x}_T$ is approximately Gaussian noise, and at $\tau = 0$, $\mathbf{x}_0$ corresponds to clean audio. The probability-flow ODE for this process is given by

$$d\mathbf{x}_\tau = -\tau \nabla_{\mathbf{x}_\tau} \log p_\tau(\mathbf{x}_\tau) d\tau, \tag{20}$$

where $\nabla_{\mathbf{x}_\tau} \log p_\tau(\mathbf{x}_\tau)$ is the prior score. We approximate this score using a denoiser network $s_\theta(\mathbf{x}_\tau, \tau)$:

$$\nabla_{\mathbf{x}_\tau} \log p_\tau(\mathbf{x}_\tau) \approx \frac{s_\theta(\mathbf{x}_\tau, \tau) - \mathbf{x}_\tau}{\sigma(\tau)^2}, \tag{21}$$

where $\sigma(\tau)^2$ is the noise variance at time $\tau$. The network is trained via denoising score matching:

$$\mathbb{E}_{\mathbf{x}_0 \sim p_{\text{data}}, \boldsymbol{\epsilon} \sim \mathcal{N}(0, \mathbf{I})} \left[ \lambda(\tau) \left\| s_\theta(\mathbf{x}_0 + \tau \boldsymbol{\epsilon}, \tau) - \mathbf{x}_0 \right\|_2^2 \right], \tag{22}$$

with $\lambda(\tau)$ chosen as in Karras et al. (2022).

## D.2 METHODS

As described in the main text, for inverse problems with observations $\mathbf{y}$, we replace the prior score in equation 20 with the posterior score $\nabla_{\mathbf{x}_\tau} \log p_\tau(\mathbf{x}_\tau \,|\, \mathbf{y})$. Using a Bayesian decomposition Chung et al. (2022), we split the posterior score at time $\tau$ as

$$\nabla_{\mathbf{x}_\tau} \log p_\tau(\mathbf{x}_\tau \,|\, \mathbf{y}) = \nabla_{\mathbf{x}_\tau} \log p_\tau(\mathbf{x}_\tau) + \nabla_{\mathbf{x}_\tau} \log p_\tau(\mathbf{y} \,|\, \mathbf{x}_\tau), \tag{23}$$

where the first term is the prior score from equation 21 and the second term is a likelihood term. In prior work Moliner et al. (2024), the likelihood score is approximated as the gradient of an audio-domain cost function $C_{\text{audio}}(\mathbf{y}, H(\hat{\mathbf{x}}_0))$, scaled by a normalization factor $\xi(\tau)$:

$$\nabla_{\mathbf{x}_\tau} \log p_\tau(\mathbf{y} \,|\, \mathbf{x}_\tau) \approx -\xi(\tau) \nabla_{\mathbf{x}_\tau} C_{\text{audio}}\big(\mathbf{y}, H(\hat{\mathbf{x}}_0)\big), \qquad \hat{\mathbf{x}}_0 = D_\theta(\mathbf{x}_\tau, \tau), \tag{24}$$

where $H(\cdot)$ denotes the (known or learned) degradation operator (e.g., a frequency-domain equalizer in BABE2). Substituting equation 23 and equation 24 into the ODE equation 20 yields the posterior-flow:

$$d\mathbf{x}_\tau = -\tau \Big[ \underbrace{\nabla_{\mathbf{x}_\tau} \log p_\tau(\mathbf{x}_\tau)}_{\text{prior}} + \underbrace{\nabla_{\mathbf{x}_\tau} \log p_\tau(\mathbf{y} \,|\, \mathbf{x}_\tau)}_{\text{likelihood}} \Big] d\tau. \tag{25}$$

Moliner et al. (2024) uses a carefully designed approximation of the $\mathcal{A}$ operator and optimizes this quantity while carrying out diffusion inference steps. We sidestep this process via our contrastive framework.

**Contrastive guidance.** To incorporate CoGuide, we add a contrastive guidance term that measures alignment between the reconstruction and the observed degraded audio in an embedding space. We use an encoder, $f_\varphi : \mathcal{X} \cup \mathcal{Y} \to \mathbb{R}^d$, to map a candidate clean signal and the corresponding observation into a shared latent space. At diffusion time $\tau$, we obtain a current reconstruction $\hat{\mathbf{x}}_0 = s_\theta(\mathbf{x}_\tau, \tau)$ and compute embeddings $f_\varphi(\hat{\mathbf{x}}_0), f_\varphi(\mathbf{y})$. We define a contrastive score via the inner product $\langle f_\varphi(\hat{\mathbf{x}}_0), f_\psi(\mathbf{y}) \rangle$, with $\ell_2$-normalized embeddings to compute cosine similarity.

During sampling, this score acts as a surrogate likelihood (like in **CoGuide**): candidates whose embeddings align with the observation are favored. Concretely, we replace the likelihood term with a contrastive gradient given by:

$$\widehat{\nabla}_{\mathbf{x}_{\tau_k}} \log p_{\tau_k}(\mathbf{y} \,|\, \mathbf{x}_{\tau_k}) \approx \gamma(\tau_k) \nabla_{\mathbf{x}_{\tau_k}} \langle f_\varphi(\hat{\mathbf{x}}_0), f_\varphi(\mathbf{y}) \rangle, \tag{26}$$

where $\gamma(\tau)$ controls the guidance strength. The resulting CoGuide update becomes

$$\mathbf{x}_{\tau_{k-1}} = \mathbf{x}_{\tau_k} - h_k \tau_k \Big[ \widehat{\nabla}_{\mathbf{x}_{\tau_k}} \log p_{\tau_k}(\mathbf{x}_{\tau_k}) + \gamma(\tau_k) \nabla_{\mathbf{x}_{\tau_k}} \langle f_\varphi(\hat{\mathbf{x}}_0), f_\varphi(\mathbf{y}) \rangle \Big], \tag{27}$$

where the gradient $\nabla_{\mathbf{x}_{\tau_k}}$ is obtained by backpropagating the inner product $\langle f_\varphi(s_\theta(\mathbf{x}_{\tau_k}, \tau_k)), f_\varphi(\mathbf{y}) \rangle$ through both the denoiser $s_\theta$ and the encoder $f_\varphi$.

## D.3 TRAINING AND IMPLEMENTATION DETAILS

**Contrastive audio model.** We train a contrastive encoder to embed clean and degraded piano audio into a shared latent space. Given a clean waveform $\mathbf{x}$ and its degraded counterpart $\mathbf{y}$, both sampled at 22,050 Hz and cropped to fixed $8.35$ s segments, we first compute their complex Constant-Q Transform (CQT) Velasco et al. (2011).

$$\mathbf{X} = \text{CQT}(\mathbf{x}) \in \mathbb{C}^{F \times T}, \tag{28}$$

$$\mathbf{Y} = \text{CQT}(\mathbf{y}) \in \mathbb{C}^{F \times T}, \tag{29}$$

using 7 octaves and 64 bins per octave. Real and imaginary parts are stacked as channels and passed to the encoder $f_\varphi$. This encoder consists of a 2D CNN/ResNet backbone over the $(F, T)$ CQT grid, followed by global average pooling and a linear projection to a $d = 256$-dimensional embedding, with $\ell_2$-normalization. **Training.** Clean data are drawn from the MAESTRO dataset Hawthorne et al. (2018), and degradations are applied on-the-fly in the waveform domain (e.g., low-pass filtering, spectral coloration, clipping, additive Gaussian noise, and mild dynamic range compression), producing pairs $(\mathbf{x}_i, \mathbf{y}_i)$. For a mini-batch, we compute similarities by inner products scaled by a temperature $\tau$. We use a symmetric InfoNCE loss so

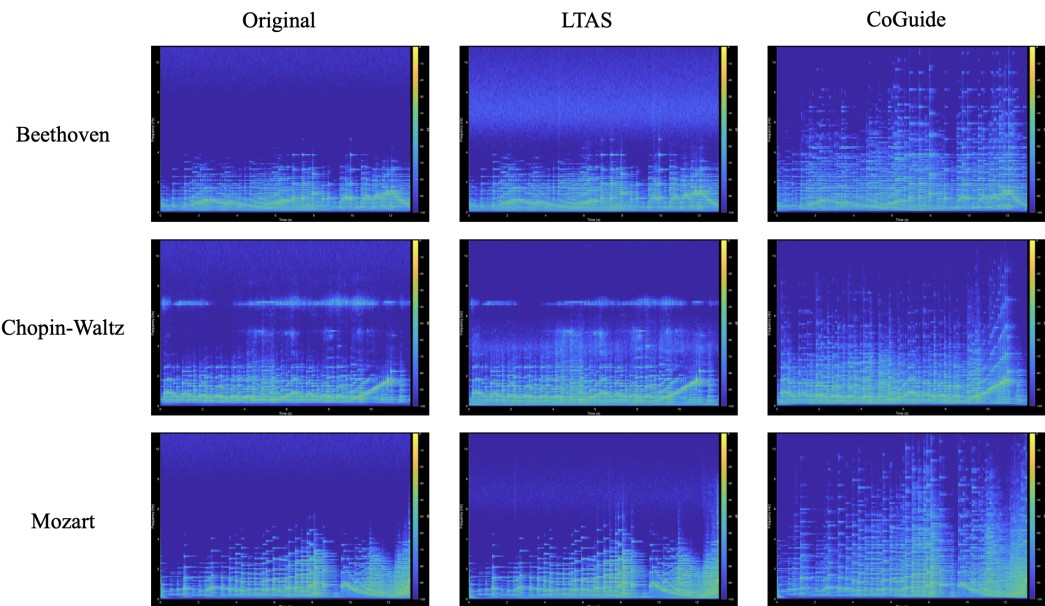

Figure 11: Comparison of spectrograms for the degraded (original), LTAS, and **CoGuide** across three different pieces of music from the MAESTRO dataset Hawthorne et al. (2018).1

that each clean embedding treats its own degraded counterpart as the positive and all others as negatives (and vice versa). For a batch of size $B$, the loss is

$$\mathcal{L} = \frac{1}{2B}\sum_{i=1}^{B}\left[-\log\frac{\exp\big(\langle f_\varphi(\mathbf{X}_i), f_\varphi(\mathbf{Y}_i)\rangle/\tau\big)}{\sum_{j=1}^{B}\exp\big(\langle f_\varphi(\mathbf{X}_i), f_\varphi(\mathbf{Y}_j)\rangle/\tau\big)} - \log\frac{\exp\big(\langle f_\varphi(\mathbf{Y}_i), f_\varphi(\mathbf{X}_i)\rangle/\tau\big)}{\sum_{j=1}^{B}\exp\big(\langle f_\varphi(\mathbf{Y}_i), f_\varphi(\mathbf{X}_j)\rangle/\tau\big)}\right]$$

The model is trained with AdamW (learning rate $10^{-4}$, weight decay 0), batch size $B = 256$, and a fixed temperature $\tau = 0.07$, until the validation contrastive loss and embedding alignment metrics saturate. The trained encoder $f_\varphi$ is then frozen and used to provide contrastive guidance during diffusion inference.

### D.4 RESULTS

Fig. 11 shows qualitative audio samples with spectrograms below for comparison across different composers' pieces. **CoGuide** is able to reduce nonlinear effects, perform denoising of nonstationary noise, and enable the reconstruction of high-frequency details. Table 4 presents the quantitative metrics using Fréchet Audio Distance (FAD) Kilgour et al. (2018) (computed with VGGNet and PANN Kong et al. (2020a)) for the degraded recordings, the LTAS baseline, and **CoGuide**. Our method outperforms the baseline in both metrics. The LTAS baseline Moliner et al. (2024) is an equalization technique that seeks to align the Long-term Averaged Spectrum (LTAS) of the audio with a reference set. Additional qualitative audio samples are available at our project page.

| Audio Type | VGG ↓ | PANN ↓ |
|---|---|---|
| Degraded (Original Recording) | 2.52 | 0.39 |
| LTAS Baseline | 2.88 | 0.27 |
| **CoGuide** restored (Ours) | **0.84** | **0.19** |

Table 4: Quantitative FAD comparison of **CoGuide** against the original recording and LTAS baseline for blind audio enhancement. FAD-PANN results are reported after scaling by $10^3$.

# E    Path-planning algorithms

We now briefly discuss the planning algorithms and their implementation details used in our experiments.

## E.1    AStar (A*)

AStar is a classical path-planning algorithm that computes the shortest path on a grid maze given start and goal locations. A* has been used in robotics and game development for navigation tasks. To compute the shortest path, A* maintains two lists: an open list of nodes $\mathcal{O}$ to be evaluated and a closed list of nodes $\mathcal{C}$ already evaluated. At each step, among the open list $\mathcal{O}$, A* selects the node with the lowest cost $f(n) = g(n) + h(n)$, where $g(n)$ is the cost from the start node to node $n$, and $h(n)$ is a heuristic estimate from $n$ to the goal. Once a node is selected, it is moved to the closed list, and its neighbors are evaluated (expanded and added to the open list if they are not already in the closed list). The process continues until the goal node is reached or the open list is empty (goal not reachable). The heuristic $h(n)$ is typically chosen to be admissible, meaning it never overestimates the true cost to reach the goal. Typically the Euclidean, Manhattan distance or Octile distance is chosen as the heuristic $h(n)$. A* is complete and optimal, meaning it is guaranteed to find the shortest path if one exists and the heuristic is admissible. In our problem ,we recast A* as a forward operator $\mathcal{A}(.)$ that takes in a floorplan $\mathbf{x}$, start postion and an end position, and outputs the trajectory between the start and goal locations, trajectory $= \mathcal{A}(\mathbf{x}, \text{start}, \text{end})$. This emulates a human/robot navigation model where the agent always takes the shortest path between two locations. Because A* relies on node selection of minimum cost and discrete expansions, it is inherently non-differentiable and hence cannot be used in gradient-based optimization.

## E.2    Neural AStar (NA*)

To make the A* search differentiable, Neural A* (NA*) Yonetani et al. (2021) was proposed. NA* first encodes the problem instance (floorplan, start, goal) using a convolutional neural network to extract a guidance map. This guidance map learns to effectively highlight regions of the floorplan that are likely to be part of the optimal path. Then an iteratively *differentiable search* mechanism is employed on the guidance map to compute the search histories. NA* redesigns nodes in the open and closed lists $\mathcal{O}, \mathcal{C}$ as matrices and replaces the argmin operation with a softmin operation for the backward pass. Along with some clever node expansions and updates, the search process becomes differentiable. Once the goal is reached, a backtracking step is performed to extract the optimal path from the search histories. It is important to note that the end-to-end planning of NA* is still non-differentiable because of the backtracking step. So the optimization can be carried only until the "histories" step in the NA* algorithm. Finally, note that the only "learnable" component in NA* is the convolutional neural network that generates the guidance map and not the differentiable search process itself. Recall that as shown in Fig 1 (Right), a small hole opening in a wall can cause a large change in the trajectory generated by A* as the path can now go through the wall instead of going around it. This behavior would make the histories very sensitive to small changes in the environment, which would make gradient-based optimization unstable.

## E.3    Transpath

Transpath Kirilenko et al. (2023) is another differentiable path-planning algorithm that is proposed to accelerate the path-planning. By leveraging the instance-dependent structure (for example, floorplan, start, stop), Transpath learns a heuristic function using a transformer architecture Vaswani et al. (2017) to guide the search process. A path probability map (PPM) is generated from the transformer that highlights regions of the floorplan that are likely to be part of the optimal path. This PPM is then accompanied by either a focal search or a greedy search that generates the final trajectory. Hence, similar to NA*, Transpath is also not fully differentiable due to the non-differentiable nature of this search process. But we find that the PPM generated from the transformer is a good approximation of the trajectory and can be used as a proxy for the trajectory in gradient-based optimization. In all our experiments, we refer to the PPM generated from the transformer as the output of Transpath to maintain differentiability. We use the official implementation of Transpath from Kirilenko et al. (2023) and use their checkpoints for evaluation.

### E.4 DIPPER

More recently, DiPPeR Liu et al. (2024) was proposed as a diffusion-based path-planning algorithm. DiPPeR formulates path planning as a conditional generation problem where the goal is to generate a trajectory given a floorplan, start and goal locations. As official code was unavailable, we implemented DiPPeR using a U-Net architecture similar to DDPM Ho et al. (2020) to model the diffusion process directly on the trajectory space instead of the image pixel space. The U-Net takes as input a noisy trajectory $\mathbf{x}_t \in \mathbb{R}^{L \times 2}$, floorplan $\in \mathbb{R}^{H \times W}$, start $\in \mathbb{R}^{H \times W}$ and goal $\in \mathbb{R}^{H \times W}$, and time step $t$ and outputs the denoised trajectory. DiPPeR is trained using a standard denoising loss using A* generated trajectories as ground truth. At inference, DiPPeR starts from a random noisy trajectory and iteratively denoises it to generate a valid trajectory. Because the model has to learn "intersections" implicitly, DiPPeR sometimes struggles with complex environments in producing intersecting trajectories that are not valid. To see this, observe the last column in Fig 12 where the DiPPeR is run over 5 different seeds. While the first two rows show non-intersecting paths, the remaining bottom three have small intersections, which would not happen for traditional planners. Although DiPPeR is differentiable end-to-end, the denoising process is computationally expensive as it requires multiple forward passes through the U-Net to get one gradient step. Accelerated sampling methods such as DDIM Song et al. (2020a) can be used to reduce the number of forward passes, and we use 50 DDIM steps in our experiments. This explicit intersection-unawareness makes DiPPeR less suitable as a proxy forward operator in our setting.

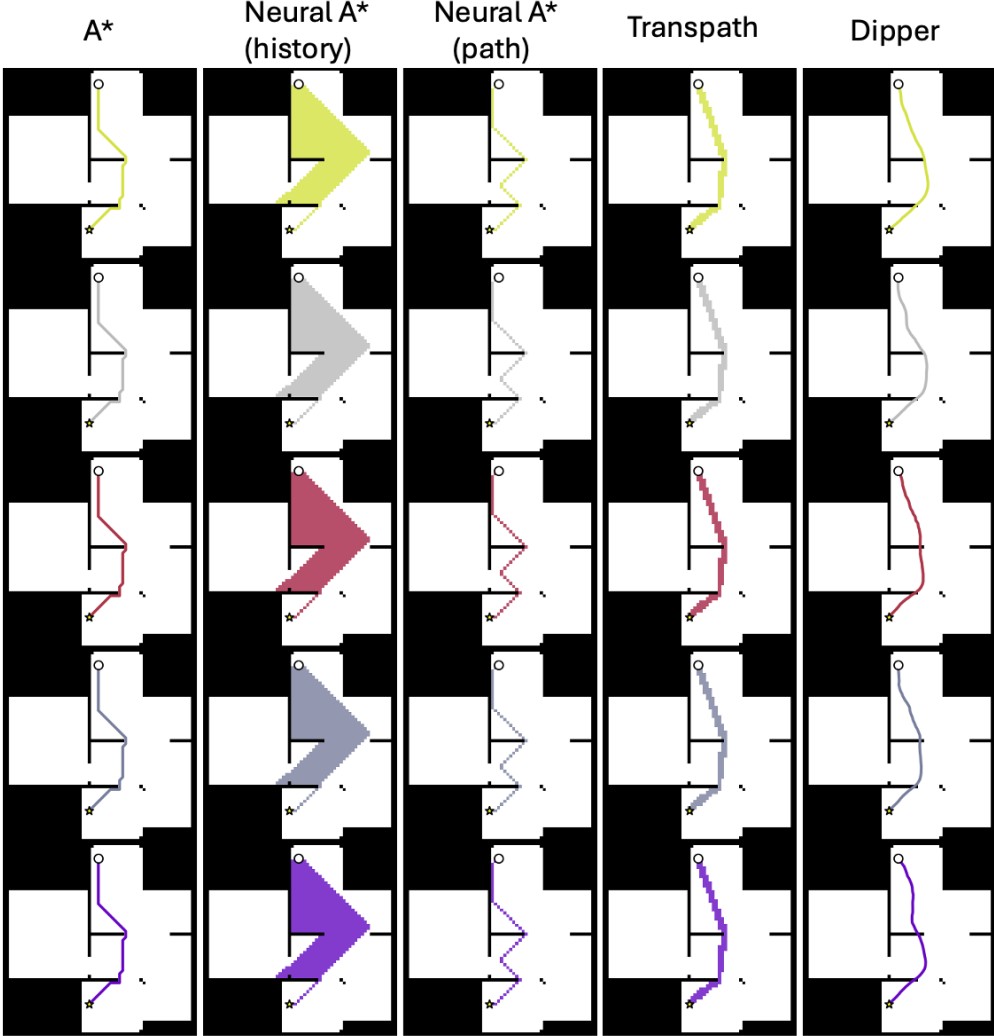

Figure 12: Comparison of DiPPeR across seeds.

## E.5 COMPARISON OF PLANNERS

We compare the behavior of A*, NA*, TransPath, and DiPPeR from the same start and end on multiple floorplans to analyze their performance. We note that all the planners are able to plan paths well in these binary (black walls and free spaces). Also, as mentioned in E.1, only the history of Neural A* is differentiable.

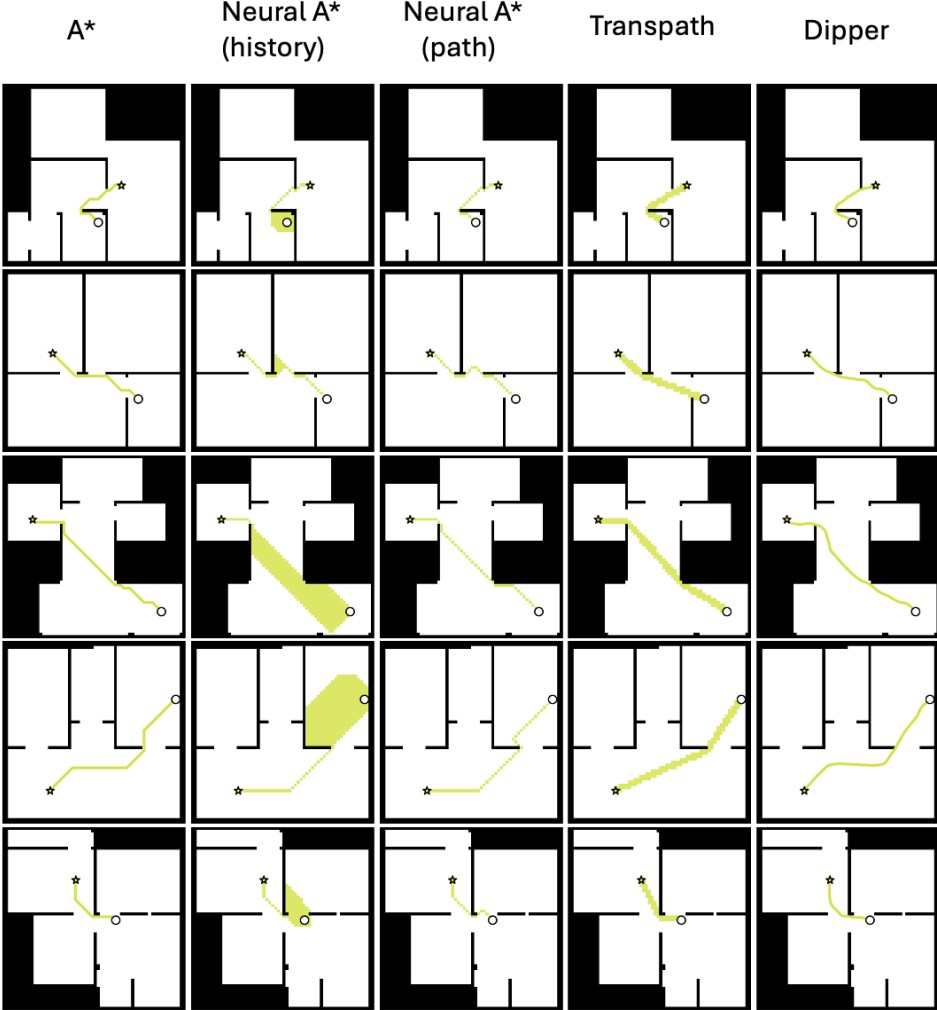

Figure 13: Comparison of various path planners.

