# OpenReview forum: "Contrastive Diffusion Guidance for Spatial Inverse Problems"
_ICLR.cc/2026/Conference — ICLR 2026 Poster_

### Official Review · Reviewer_cV1B · 2025-10-30

**Soundness:** 3
**Presentation:** 3
**Contribution:** 3
**Rating:** 6
**Confidence:** 3

**Summary:**

This paper addresses the spatial inverse problem of reconstructing 2D floorplans from human trajectories. The proposed method, CoGuide, is a diffusion-based posterior sampler that avoids the instability of path-planning forward operators by learning a smooth, shared embedding space for floorplans and trajectories using contrastive learning. The learned contrastive similarity score functions as a stable surrogate for the true likelihood, steering the reverse diffusion process toward a compatible layout. CoGuide was evaluated on reconstructing $64\times64$ floorplan images from synthetic trajectories. Experiments show the method produces more robust and visually consistent floorplans than baselines using differentiable planners (e.g., DPS+Neural A*) and achieves the best overall performance.

**Strengths:**

- The paper clearly motivates its approach by demonstrating that the core path-planning forward operator is unstable, non-differentiable, and non-smooth.

- The central proposal to learn a smooth likelihood surrogate in a contrastive embedding space is a novel and well-reasoned solution to bypass the problematic forward operator of the studied problem.

- The experiments are thorough, comparing CoGuide against six relevant baselines (including DPS+planners and CFG) and showing superior results in sparse and moderate data regimes.

**Weaknesses:**

- The method might suffer from a significant "sim-to-real" gap, as the encoders are trained exclusively on synthetic $A^*$ (shortest-path) trajectories and on empty floorplans lacking real-world obstacles like furniture. The authors should discuss how the approach would generalize to more complicated real-world problem settings.

- There is no explanation for why the CFG baseline outperforms CoGuide in dense trajectory settings.

- The method's practical applicability is questionable due to scalability, as it is only demonstrated on small $64\times64$ images, a limitation the authors attribute to the poor scaling of path planners.

**Questions:**

- The use of an Adam optimizer inside each DDIM guidance step is an unusual modification; what are the theoretical implications of this on the mathematical validity and convergence properties of the reverse diffusion process? Why not use other higher-order ODE solvers?

- How sensitive is the model to the choice of the synthetic data generator? Would performance degrade significantly if the trained models are directly tested with real-world human trajectories?

---

> ### Author Response · Authors · 2025-11-25
>
> > __Q1. Significant sim-to-real gap due to synthetic trajectories and empty floorplans. How would this generalize to real-world settings? Similar comments from reviewers 8553, 7FMZ__
>
> We acknowledge that CoGuide has a sim2real gap in terms of the trajectories. To close this gap, we collected real-world trajectories (collected via ultrawideband (UWB) wireless sensors described below) in 2 student apartments. These trajectories are relatively sparse, they go around furniture and are inaccurate due to measurement errors. Importantly, these together model human trajectories in the real-world, different from the Neural A* distribution.
>
> We applied CoGuide on these trajectories and compared against CFG. New results are now reported in the paper (Figure 5) — although CoGuide estimated floorplans have mistakes, we observe that several walls have been positioned correctly, and our results visibly outperform CFG. If the contrastive model were trained with such real-world human trajectories and furniture-embedded floorplans, we believe results could be substantially better, narrowing the sim2real gap.
>
> Implementation details: Real-world measurement process using UWB sensors (included in Section 4).
>
> We collected walking trajectory data from 2 student apartments, for which ground-truth floorplans were obtained from the leasing company websites.
> We collected trajectories using an Ultrawideband (UWB) localization system based on the DWM3000 module. This provides the 2D location of a user with an accuracy of 10cm standard deviation as they walk within the household. The UWB system was chosen because of its ability to provide ground truth trajectory measurements, which can then be perturbed by various sources of noise to more accurately simulate a real-world deployment.
>
> More specifically, the UWB system consists of multiple anchors statically placed at known 3D coordinates within a household. The user is provided a beacon that periodically pings each anchor, to estimate the Time-of-Flight (ToF) to that anchor. By utilizing this ToF to an anchor that is in the line-of-sight of the beacon and the speed of light, we can use trilateration to estimate the user's 3D position. This is then projected onto the 2D map to generate trajectories as the user walks within the household.

---

> ### Author Response · Authors · 2025-11-25
>
> >__Q2. Why do CFG baselines outperform CoGuide in dense trajectory settings?__
>
> We understand the question. CoGuide outperforms CFG in sparse and medium trajectory settings, but in the dense case, we think the many trajectories give abundant conditioning information for CFG to overfit to the joint trajectory-floorplan distribution. In fact, CFG may be able to solve many inverse problems by training on paired data, but prior work on inverse problems typically do not report comparisons with CFG. We included CFG in our baselines for a thorough assessment of the whole solution landscape.
>
> Moreover, to better understand CFG and CoGuide’s relative merits, we have now performed a new experiment that compares a CFG+CoGuide model against individual CFG and CoGuide models. The goal is to test if CoGuide captures new information beyond what CFG has learnt. For CFG+CoGuide, we replaced our unconditional diffusion prior with the CFG-trained conditional prior. Results (see table below) show distinct gains, indicating that our contrastive guidance offers expressive likelihood-based information to the denoising process that is otherwise not captured by CFG.
>
> | Method         | Sparse        |              | Moderate       |              | Dense          |              |
> |----------------|---------------|--------------|----------------|--------------|----------------|--------------|
> |                | F1            | IoU          | F1             | IoU          | F1             | IoU          |
> | CFG            | 0.86 ± 0.06   | 0.76 ± 0.10  | 0.93 ± 0.03    | 0.88 ± 0.05  | 0.97 ± 0.01    | 0.95 ± 0.03  |
> | CoGuide        | 0.91 ± 0.04   | 0.84 ± 0.07  | 0.94 ± 0.03    | 0.89 ± 0.05  | 0.95 ± 0.03    | 0.90 ± 0.06  |
> | **CFG+CoGuide**| **0.93 ± 0.04** | **0.87 ± 0.07** | **0.97 ± 0.03** | **0.93 ± 0.04** | **0.99 ± 0.01** | **0.97 ± 0.02** |
>
> &nbsp;
>
> >__Q3. Practical applicability is questionable due to scalability and small images. Similar comments from reviewers 8553, 7FMZ, rYah__
>
> Our method is not fundamentally tied to small image size. We chose this size specifically since our baselines use path-planners as forward operators. Running path-planning algorithms on large images becomes prohibitively slow when creating datasets, and also during the diffusion inference process. Using 64x64 images mitigates these problems. In response to reviewer 8553, we have now shown generalization to the audio blind inverse problems where we used CQT spectrograms of size 480x1980.
>
> &nbsp;
>
> >__Q4. Why use ADAM inside each DDIM guidance step? Why not use other higher-order solvers? Similar comments from reviewer 8553__
>
> We view Diffusion Posterior Sampling as maximizing an optimization objective (the likelihood) with regularization (the diffusion prior). In this context, our use of Adam inside each DDIM is best viewed as a numerical optimization technique, rather than as a modification of the underlying reverse SDE/ODE. Higher-order ODE samplers address a different problem than Adam and are not a replacement. Methods such as Heun or Runge–Kutta assume a fixed drift ($f(x_t, t)$) and aim to approximate the time evolution $(\frac{dx_t}{dt} = f(x_t, t))$ more accurately in $(\Delta t)$, by combining multiple **instantaneous** evaluations of the score within a single step. Their “second-order” nature is in the time-integration sense and they do not maintain any gradient history or adapt per-coordinate step sizes.
>
> In contrast, Adam is a first-order optimizer with exponential moving averages of past gradients and squared gradients. The squared gradients provide approximate second-order curvature-guidance which helps with better convergence with DDIM in our highly multi-modal posterior landscape in a fewer steps compared to DDPM.
>
> Tackling the exact mathematical implications of using Adam in the posterior sampling setting is an interesting research direction. We feel it is relevant to the next round of improvements on CoGuide and we hope to pursue it in the future.

---

> ### Comment · Reviewer_cV1B · 2025-11-27
>
> The authors made extensive efforts in the rebuttal to improve the paper's quality:
>
> - The sim-to-real experiments are very interesting and strengthen the potential value of the approach.
> - New results on audio inverse problems justify the transferability of the overall framework.
> - Thanks for the clarifications on the comparison against CFG and the use of ADAM inside DDIM steps.
>
> I have updated my score from 6 to 8, with a higher confidence.

---

### Official Review · Reviewer_rYah · 2025-10-31

**Soundness:** 3
**Presentation:** 4
**Contribution:** 3
**Rating:** 8
**Confidence:** 2

**Summary:**

This paper introduces CoGuide, a diffusion based approach to solve the inverse problem of reconstructing a floorplan from a user's movement trajectories. A key challenge of the approach is that the forward operator in a floor plan is non-differentiable and non-linear, which destabilizes standard diffusion-based solvers that rely on a likelihood score. CoGuide's key contribution is to bypass this issue by reformulating the likelihood score in a smooth, learned embedding space. This space is trained using a contrastive loss that pulls compatible floorplan-trajectory pairs together while pushing mismatched pairs apart. This creates a stable surrogate likelihood score that can effectively guide the diffusion process, enabling the model to produce more consistent and robust floor plans than baseline methods.

**Strengths:**

1. The work is well written and easy to follow. With enough preliminaries, backgrounds and technical details, it easy for readers not familiar with spatial inverse problems to understand the problem setting and the work's contributions.
2. Each component of the proposed approach is well-motivated, including the learning of the embedding space using contrastive learning, adapting and improving the diffusion models for the spatial inverse problems. The choice of these components are also well supported by ablation study results.
3. The quantitative evaluation results of the work demonstrates competitive performance of the proposed approach against various baseline models.

**Weaknesses:**

My biggest concern of the approach is on its comparison against the classifier free guidance approach. The performance gap between CFG and COGUIDE is not significant enough in either sparse, moderate, or dense setting. Is the performance gap a results of randomness or hyper-parameter selection? There're not enough details on the exact instantiation of the baseline CFG model which is also a diffusion-based model and its difference with the proposed COGUIDE approach. More ablation study focusing on the difference between CFG and COGUIDE might help with answer this question.

**Questions:**

Floorpans are downsampled to a resolution of 64x64 in the work's experiment setting. Is this a common practice in solving spatial inverse problem setups? Is there any challenge preventing the approach from being applied to higher resolution inputs?

---

> ### Author Response · Authors · 2025-11-25
>
> >__Q1. More ablation studies focussing on the difference between CFG and CoGuide.
> Similar comments from reviewers  7FMZ__
>
> This is a pertinent question and we have thought about this ourselves.
>
> CoGuide outperforms CFG in sparse and medium trajectory settings, but in the dense case, we think the many trajectories give abundant conditioning information for CFG to overfit to the joint trajectory-floorplan distribution. In fact, CFG may be able to solve many inverse problems by training on paired data, but prior work on inverse problems typically do not report comparisons with CFG. We included CFG in our baselines for a thorough assessment of the whole solution landscape.
>
> Moreover, to better understand CFG and CoGuide’s relative merits, we have now performed a new experiment that compares a CFG+CoGuide model against individual CFG and CoGuide models. The goal is to test if CoGuide captures new information beyond what CFG has learnt. For CFG+CoGuide, we replaced our unconditional diffusion prior with the CFG-trained conditional prior (see model details below). Results show distinct gains, indicating that our contrastive guidance offers expressive likelihood-based information to the denoising process that is otherwise not captured by CFG.
>
> | Method         | Sparse        |              | Moderate       |              | Dense          |              |
> |----------------|---------------|--------------|----------------|--------------|----------------|--------------|
> |                | F1            | IoU          | F1             | IoU          | F1             | IoU          |
> | CFG            | 0.86 ± 0.06   | 0.76 ± 0.10  | 0.93 ± 0.03    | 0.88 ± 0.05  | 0.97 ± 0.01    | 0.95 ± 0.03  |
> | CoGuide        | 0.91 ± 0.04   | 0.84 ± 0.07  | 0.94 ± 0.03    | 0.89 ± 0.05  | 0.95 ± 0.03    | 0.90 ± 0.06  |
> | **CFG+CoGuide**| **0.93 ± 0.04** | **0.87 ± 0.07** | **0.97 ± 0.03** | **0.93 ± 0.04** | **0.99 ± 0.01** | **0.97 ± 0.02** |
>
> As a final comparison against CFG, we have also performed real-world experiments using human-walked trajectories, measured in 2 student apartments. Results show that CoGuide performs appreciably better than CFG (see a new Fig. 5 in the main paper).

---

> ### Author Response · Authors · 2025-11-25
>
> > __Q2. More details needed on the exact instantiation of the baseline CFG and the CoGuide model.__
>
> Thanks for pointing this out. We have summarized them below and have included all details of the CFG model, including implementation, training details, and hyperparameter settings, in  Appendix C.2.
>
> The model uses a UNet architecture with four resolution stages, beginning at 64×64 and expanding its channel capacity across levels (128, 256, 384, and 512). Each stage contains a single residual block, and the UNet includes a standard mid-block before the upsampling path mirrors the downsampling hierarchy. Group normalization with 32 groups and scale-shift time embeddings are applied throughout. The network is trained for 50 epochs on 64×64 images using a batch size of 128, a learning rate of 1e-4 with 500 warmup steps, and no gradient accumulation. Classifier-free guidance is enabled through a 0.1 dropout probability during training and a guidance scale of 1.5 at inference. Training is capped at 50 epochs because the validation curve plateaus, indicating that further optimization does not yield additional improvement.
>
> > __Q3. Floorplans are downsampled to 64x64 resolution. Why not apply to higher resolution? Similar comments from reviewers  8553, cV1B, 7FMZ__
>
> Our method is not fundamentally tied to small image size. We chose this size specifically since our baselines use path-planners as forward operators. Running path-planning algorithms on large images becomes prohibitively slow when creating datasets, and also during the diffusion inference process. Using 64x64 images mitigates these problems. In response to reviewer 8553, we have now shown generalization to the audio blind inverse problems (new demos available at https://coguide.github.io/ and details are included in Appendix D)  where we used CQT spectrograms of size 480x1980.

---

> ### Comment · Reviewer_rYah · 2025-11-28
> **Post-Rebuttal Acknowledgement**
>
> The reviewer has addressed all my concerns and I will keep my positive rating for the work.

---

### Official Review · Reviewer_7FMZ · 2025-10-31

**Soundness:** 3
**Presentation:** 3
**Contribution:** 3
**Rating:** 6
**Confidence:** 1

**Summary:**

This paper proposes CoGuide to infer a floorplan from human-walked trajectories. It learns a joint contrastive embedding for trajectory and floorplan pairs and uses the embedding similarity as a likelihood surrogate to guide diffusion sampling. Practical additions include a trajectory–wall intersection penalty and DDIM steps optimized with Adam + cosine LR + early stop. On HouseExpo, CoGuide beats the baselines under sparse–moderate trajectories and is comparable to CFG when trajectories are dense.

**Strengths:**

Clear derivation linking InfoNCE logits to a likelihood ratio and gradients to the likelihood score; switching to unit-norm embeddings yields an equivalent squared-distance form that is smooth for guidance.
Using Adam within each step, cosine LR ramp + hard gating, and a trajectory–wall intersection penalty are simple, effective, and empirically ablated.
The paper diagnoses instability from 𝐴(⋅), which has large/sensitive Jacobians, as well as non-smooth choices, motivating the embedding approach.

**Weaknesses:**

Evaluation is entirely synthetic on 64×64 binary maps, which makes the sim2real validity uncertain.

While baselines are diverse, CFG wins in dense trajectories. It would be fair to analyze why and whether CFG + contrastive guidance is helpful. Additionally, consider stronger conditional diffusion variants, such as those with modern UNets or latent-diffusion backbones.

Only Gaussian perturbations to trajectories are tested. Yey, real sensors have biases and drift. A more realistic corruption suite would be convincing.

**Questions:**

Any results at higher resolution or with grayscale walls/soft obstacles to test robustness beyond binary maps?

How does performance change with real trajectories, e.g., human teleop or logs from indoor SLAM?

Can you combine CoGuide with CFG to close the dense-trajectory gap?

---

> ### Author Response · Authors · 2025-11-25
>
> >__Q1. Analyze whether CFG + Contrastive guidance is helpful. Can you combine CFG + CoGuide to close the dense-trajectory gap? Why not use modern UNets?
> Similar comments from reviewers cV1B, rYah__
>
> We thank the reviewer for this constructive suggestion. It is indeed a valuable idea to combine CFG and CoGuide so they can complement each other’s strengths. In our implementation, CFG serves as a conditional prior over floorplans. Using this setup, we have re-run experiments across sparse, medium, and dense trajectory regimes; the corresponding results are shown in the table below (and implementation details are included in the Appendix C.2).
>
> Results show that the combined model consistently outperforms each of the individual models (CFG and CoGuide) across all levels of trajectory sparsity. Since CFG was already our strongest baselines, these additional gains can be attributed to CoGuide’s ability to further tighten the likelihood term and improve inference. This modification is simple: the only change is to replace CoGuide’s original diffusion prior with the CFG model. When desired, the same CFG prior can be given null inputs $\varnothing$, recovering the original CoGuide behavior. If the paper is accepted, we plan to include this combined model and its results in the final version.
>
> Regarding modern architectures, it is perhaps true that newer diffusion backbones (ViTs, Latent-diffusion, etc.) would yield stronger performance. However, we wanted to highlight the core conceptual benefits of contrastive guidance, for which we believe our current UNet based diffusion prior is a good candidate.
>
> ---
>
>
> | Method         | Sparse        |              | Moderate       |              | Dense          |              |
> |----------------|---------------|--------------|----------------|--------------|----------------|--------------|
> |                | F1            | IoU          | F1             | IoU          | F1             | IoU          |
> | CFG            | 0.86 ± 0.06   | 0.76 ± 0.10  | 0.93 ± 0.03    | 0.88 ± 0.05  | 0.97 ± 0.01    | 0.95 ± 0.03  |
> | CoGuide        | 0.91 ± 0.04   | 0.84 ± 0.07  | 0.94 ± 0.03    | 0.89 ± 0.05  | 0.95 ± 0.03    | 0.90 ± 0.06  |
> | **CFG+CoGuide**| **0.93 ± 0.04** | **0.87 ± 0.07** | **0.97 ± 0.03** | **0.93 ± 0.04** | **0.99 ± 0.01** | **0.97 ± 0.02** |

---

> ### Author Response · Authors · 2025-11-25
>
> >__Q2. Evaluation is synthetic on 64x64 binary maps, making sim2real validity uncertain.
> Similar comments from reviewers 8553, cV1B, rYa__
>
> Our method is not fundamentally tied to small image size. We chose this size specifically since our baselines use path-planners as forward operators. Running path-planning algorithms on large images becomes prohibitively slow when creating datasets, and also during the diffusion inference process. Using 64x64 images mitigates these problems. In response to reviewer 8553, we have now shown generalization to the audio blind inverse problems where we used CQT spectrograms of size 480x1980.
>
> >__Q3. What about grayscale representations for walls/soft obstacles to test beyond binary images.
> Similar comments from reviewers 8553, cV1B__
>
> For path-planning, a black pixel indicates a wall or an obstacle that disallows any path from going through that pixel; a white pixel indicates an empty space allowing a path to traverse through. With grayscale images, this behavior becomes unclear. This is the reason path-planning operations [1,2] are applied on binary images and we have followed that approach.
>
> We agree that it should be possible to apply CoGuide to more detailed floorplans (including furniture) should such datasets become publicly available.
>
> __Reference:__
>
> [1] Dipper: Diffusion-based 2d path planner applied on legged robots. Liu, J. et al., ICRA 2024
>
> [2] Transpath: Learning heuristics for grid-based pathfinding via transformers. Kirilenko, D, et al. AAAI 2023

---

> ### Author Response · Authors · 2025-11-25
>
> > __Q4. Performance with real trajectories (e.g., human teleop, indoor SLAM, and real sensor perturbations)
> Similar comments from reviewers 8553, cV1B__
>
> We understand that A* and Neural A* operators do not model human behavior. To close this gap, we collected real-world trajectories (collected via ultrawideband (UWB) wireless sensors described below) in 2 student apartments. These trajectories are relatively sparse, they go around furniture and are inaccurate due to measurement errors. Importantly, these together model human trajectories in the real-world, different from the Neural A* distribution.
>
> We applied CoGuide on these trajectories and compared against CFG. New results are now reported in the paper (Figure 5) — although CoGuide estimated floorplans have mistakes, we observe that several walls have been positioned correctly, and our results visibly outperform CFG. If the contrastive model were trained with such real-world human trajectories and furniture-embedded floorplans, we believe results could be substantially better, narrowing the sim2real gap.
>
> Implementation details: Real-world measurement process using UWB sensors (included in Section 4).
>
> We collected walking trajectory data from 2 student apartments, for which ground-truth floorplans were obtained from the leasing company websites.
> We collected trajectories using an Ultrawideband (UWB) localization system based on the DWM3000 module. This provides the 2D location of a user with an accuracy of 10cm standard deviation as they walk within the household. The UWB system was chosen because of its ability to provide ground truth trajectory measurements, which can then be perturbed by various sources of noise to more accurately simulate a real-world deployment.
>
> More specifically, the UWB system consists of multiple anchors statically placed at known 3D coordinates within a household. The user is provided a beacon that periodically pings each anchor, to estimate the Time-of-Flight (ToF) to that anchor. By utilizing this ToF to an anchor that is in the line-of-sight of the beacon and the speed of light, we can use trilateration to estimate the user's 3D position. This is then projected onto the 2D map to generate trajectories as the user walks within the household.

---

> > ### Comment · Reviewer_7FMZ · 2025-11-28
> >
> > Thanks for the response. I am not entirely familiar with this area. I would like to keep positive and let the decisions be made by other expert reviewers and the AC.

---

### Official Review · Reviewer_8553 · 2025-11-02

**Soundness:** 2
**Presentation:** 3
**Contribution:** 3
**Rating:** 4
**Confidence:** 4

**Summary:**

The paper proposes CoGuide, a diffusion-based method for spatial inverse problems with the example in the paper being room layout estimation from (simulated) human trajectories. The authors formalize this task as an inverse problem, where the trajectories have been generated by a forward process that applies an operator (in this case a path planning process by a human) on a certain room layout.
The goal now is estimate a plausable room layout from a given trajectory. For this, the authors follow the existing framework of Diffusion Posterior Sampling [1], which leverages a trained diffusion model as a data prior and additionally maximizes the likelihood of the observation from the foward process (the trajectory inside the room in this case) during the generation / reverse process, i.e., sampling of the unobserved signal (the room layout). However, for the application to room layout estimation from trajectories, the authors identify the use of differentiable path planner (A*) approximations for the likelihood term as unstable / very sensitive to small sample perturbations.
Therefore, the authors propose a new method CoGuide that trains encoders for trajectories and floor plans to a common embedding space with contrastive losses and approximates the likelihood term via similarities of embeddings. Additionally, the paper introduces an additional intersection penalty term and explores using Adam instead of SGD for the gradient-based guidance during sampling.
Finally, an experimental evaluation includes qualitative and quantitative results for different trajectory sparsity levels as well as an ablation study w.r.t. the intersection penalty and the use of Adam.

References:
- [1] Diffusion Posterior Sampling for General Noisy Inverse Problems. ICLR 2023

**Strengths:**

- The paper is very well written and easy to follow and understand.
  - The introduction does a great job in formalizing inverse problems, the application to floor plan estimation from human trajectories, and already summarizes the main idea of the paper in an understandable way.
  - Preliminaries cover the important existing works to understand the contributions of this paper.
  - The method section first formulates the problem (in particular why a straightforward use of a differentiable A* approximation in the DPS framework does not work well), also visualizes it in Fig. 1, then describes the approach well and proofs that it is a valid approximation of the true likelihood, and finally visualizes the embedding space in a t-SNE plot to give insight about why their method works intuitively.
- The method is interesting, general, and to my knowledge sound and novel.
  - Making use of InfoNCE linking contrastive learning and density estimation to define a likelihood surrogate via similarity in an embedding space learned contrastively seems elegant and also generally applicable to more problems.
- The qualitative and quantitative evaluation shows strong performance compared to a variety of baselines.
- The ablation study validates the effectiveness of the wall and trajectory intersection penalty as well as the use of Adam for gradient-based guidance.
- The appendix provides further qualitative results, implementation details, and information about used path planning algorithms.

**Weaknesses:**

- Limited evaluation:
  - Even though the method seems to be general and therefore applicable to other (spatial) inverse problems, the paper is limited to the single task of floor plan estimation from trajectories on a single dataset. The paper could be much stronger, if the authors would provide at least one more application.
  - For quantitative results, the paper only provides IoU and F1 score as metrics. If I am not mistaken, both metrics do not account for multiple possible floor plans given a trajectory. Would it be possible provide distribution-based metrics similar to FID and possibly check validity in form of no wall-trajectory intersections? At least, these would allow for multiple possible solutions instead of just comparing with the single ground truth sample.
  - The authors claim that CoGuide is "more robust than differentiable-planner baselines and guided-diffusion methods" but they only show this robustness in terms of trajectory sparseness, where CoGuide outperforms all baselines for sparse and moderate trajectory densities. Robustness in terms of the level of noise in the measurements is only analyzed for CoGuide (Fig. 4) but not for baselines.
- The paper misses to discuss the floor plan and trajectory representations:
  - I find the representation as grayscale images questionable. To this end, the authors unfortunately do not discuss whether this choice is in line with previous related works and whether there could be any alternatives.
  - The authors already mention that even though both trajectories and floor plans are binary, planners would have to "cope with continuous-valued inputs (gray pixels) during the reverse diffusion process" (line 161). If both are binary, would a discrete diffusion approach be more applicable for example?
- Concern about comparison with baselines and their selection:
  - The quantitative and qualitative results are not completely convincing in comparison with the Classifier-Free Guidance baseline. It outperforms CoGuide for dense trajectories and is pretty much on par for moderate trajectory density (Tab. 1) for the two used metrics.
    - Also, how is the guidance scale tuned? Is it the best performance possible w.r.t. this hyperparameter?
  - Since both the intersection penalty and Adam are shown to be important for strong performance in their ablations (Tab. 2 and Tab. 3), the question arises whether the baselines also make use of these improvements, which are independent of the actual CoGuide method or not.
    - To this end, the intersection penalty is a straightforward way of incorporating manual inductive bias specific to floor plan estimation.
      - If I am not mistaken, it should be possible to use the intersection penalty also in combination with CFG or not? This is important since without the intersection penalty, CFG would mostly outperform CoGuide.
    - The use of Adam for the gradient-based guidance during the reverse process seems like something completely general to me independent of the application or the rest of the CoGuide method. Is this paper really the first one to do this?

- Minor weaknesses:
  - In the method section, it could be beneficial to already show some snippet of failure cases from baselines using differentiable A* approximations for the likelihood computation as evidence for the claims and explanations that these are unstable.
  - The Sec. 3.2 (and the rest of the main paper) does not include any details about the encoder architectures.
  - It could be interesting to show t-SNE plots for a larger part of the dataset as well.
  - The GitHub links directly in text in the paragraph about baselines should be avoided. One could just say that you follow the original configurations or at most have them as footnotes.

**Questions:**

The authors should discuss the representation of floor plans and trajectories as continuous grayscale images, also w.r.t. existing works. Furthermore, I would suggest to address my concerns regarding fair baseline comparisons w.r.t. intersection penalty, Adam, and guidance scale for the Classifier-Free Guidance baselines and others. Furthermore, additional metrics that account for the existence of multiple possible floor plans matching a trajectory would be interesting as well as a comparison with baselines in terms of robustness to measurement noise.

Possibly out of scope for rebuttal: The paper would be much stronger, if it would show the generalization of CoGuide to other applications than the single, synthetic floor plan estimation from human trajectories.

---

> ### Author Response · Authors · 2025-11-25
>
> > __Q1. The paper could be much stronger, if the authors would provide at least one more application.__
>
> We want to believe that CoGuide is generalizable to other applications, but did not act on it until now (we were too focussed on the unique challenges of the path-planning application).  However, motivated by your review comments, we have now applied CoGuide to an audio enhancement application. Specifically, we aim to enhance historical piano recordings corrupted by unknown degradations (i.e., the A(.) operator includes spectral coloration, additive noise and clicks, nonlinear effects, etc.). This is a fully blind setting since we neither assume an analytical form for A(.), nor do we estimate it during inference. Instead, we employ our proposed contrastive model—trained with pairs of clean and degraded audio data—to learn powerful representations. Our clean data are recordings from the MAESTRO dataset [1] while the degradations include low-pass filtering, clipping, additive Gaussian noise, etc. Using a pretrained diffusion prior [2], we observe improved audio quality of these recordings; CQT plots demonstrate that the model achieves bandwidth extension, improved dynamics and a reduction in noise in the high frequency. We compare against LTAS [2], an equalization technique that seeks to align the Long-term Averaged Spectrum of the audio with a reference set. Fréchet Audio Distance (FAD) results and training details are provided below, and audio demos are now included on our project website at https://coguide.github.io/. Needless to say, we developed this in a short timeframe; we expect a more careful design and tuning is to boost the model’s performance.
>
>
> | Audio Type              | VGG ↓ | PANN ↓ |
> |---------------------|------------:|-------:|
> | Degraded (Original) |        2.52 |   0.39 |
> | LTAS                |        2.88 |   0.27 |
> | CoGuide (Ours)      |        0.84 |   0.19 |
>
> ---
>
> **Method and training details for audio restoration (now added in Appendix D):**
>
> - Diffusion prior:
> We used the pretrained diffusion model based on the CQTDiff+ architecture from BABE2 [2]. This model is trained on the MAESTRO dataset [1] consisting of 200 hours of piano recordings.
> - Contrastive Model:
> We train a contrastive encoder on solo piano recordings from the MAESTRO v3.0.0 dataset. Each audio clip is resampled to 22.05 kHz and randomly cropped to 8.35 s (184184 samples) to match the diffusion prior. Both the clean audio snippet $\mathbf{x}$ and a degraded version $\mathbf{y}$ (produced by a random chain of audio corruptions such as additive noise, filtering, or hard-clipping) are transformed into a complex Constant-Q Transform (CQT) representation using a Kaiser window with 7 octaves and 64 bins per octave. This yields a time–frequency representation $\mathbf{X}=CQT(\mathbf{x}),\mathbf{Y}=CQT({\mathbf{y}})$ that serves as input to the shared encoders $f_\varphi$. The encoders use a convolutional backbone operating on the real and imaginary parts of CQT followed by a small projection MLP head that maps to a low-dimensional unit normalized embedding (256 dim.). The encoders share weights, so the model learns a single audio representation space in which clean and degraded versions of the same underlying audio waveform are mapped nearby. Given a batch of $N$ pairs ${(\mathbf{x}, \mathbf{y})}$, we compute inner products
> $\langle f_\varphi (\mathbf{X}) , f_\varphi (\mathbf{Y})\rangle$
> and optimize a symmetric InfoNCE loss. We train with AdamW $(lr = 1\times10^{-4})$, small weight decay), batch size 256, and temperature $(\tau = 0.07)$, for up to 100k optimization steps. Our evaluation set consisted of 54 historical gramophone recordings of solo piano performances from the early 20th century. We have provided audio files on our website to demonstrate performance.
>
> __References:__
>
> [1] Enabling Factorized Piano Music Modeling and Generation with the MAESTRO Dataset, Hawthorne C., et al., ICLR 2019.
>
> [2] A Diffusion based Generative Equalizer for Music Restoration, Moliner E., et al., DAFx 2024.

---

> ### Author Response · Authors · 2025-11-25
>
> > __Q2. Grayscale representation is questionable. If both maps and trajectories are binary, would a discrete diffusion approach be more applicable? Similar comments from reviewers 7FMZ, cV1B, rYah__
>
> Yes, we understand the question and have thought about this ourselves during the design process. However, we stuck to our choice of representing floorplans and trajectories as grayscale images to remain compatible with standard path-planning operators (which operate on a discretized grid). In our setting, each pixel corresponds to a node in the planning graph, modeling walls or free space. This representation is standard in path planning and allows us to apply off-the-shelf planners (A*, Neural A*, TransPath, DiPPeR etc.) directly. While our contrastive encoders and diffusion prior do not intrinsically require binary floorplans or strictly binary trajectory overlays, the baselines we compare against do rely on such grid-based planners. To ensure a fair comparison, we needed to commit to this format throughout our experiments.
>
> This choice is also consistent with prior work that models indoor layouts and navigation on rasterized floorplan images. For example, path-planning and localization approaches in indoor environments commonly use occupancy or semantic maps rendered as 2D grids, and recent methods such as EchoNeRF and Supercharging Floorplan Localization with Semantic Rays operate on  layout representations that are similarly discretized.
>
> Importantly, our CoGuide framework is agnostic to the exact parameterization of the layout (vectors, graphs, etc.). Therefore, using non-image representations is a promising direction for future work rather than a fundamental limitation of our method.
>
> Regarding Discrete Diffusion: We completely agree with this comment. Discrete diffusion has been on our radar as well. However, given that the diffusion prior over floorplans produced extremely reliable results even on binary images, we chose to stick with this. Our main focus therefore was on designing a smooth likelihood surrogate. In the future, we are motivated to explore additional gains that discrete diffusion could bring to the problem.

---

> ### Author Response · Authors · 2025-11-25
>
> > __Q3. Concerns regarding fair baseline comparisons w.r.t. intersection penalty, Adam, and guidance scale for the CFG baselines and others.__
>
> We understand the concern. For the Classifier-Free Guidance (CFG) baselines, we tuned the guidance scale on a test set and found (s = 1.5) to be the best choice; we have included new results and the selection procedure in the revised version in Appendix C.2.
>
> Regarding the intersection penalty, we should have been clearer in our draft. All our diffusion-posterior baselines (DPS+X, DMPlug, DiffPIR) already use the same wall–trajectory intersection loss. This term was introduced early in the project as we tried to squeeze out performance. In these baselines, it had only a marginal effect because the likelihood is computed using highly discontinuous path-planning operator. When we moved to the contrastive formulation, the same penalty understandably became much more effective. It acts as an additional steering signal that improves floorplan inference.
>
> Applying this penalty to CFG is perhaps redundant. This is because the conditional model is explicitly trained to generate non-intersecting layouts. To verify, we implemented the intersection penalty in our CFG model and did not observe any significant improvements in the metrics.
>
> Finally, in our opinion, the use of Adam inside the diffusion posterior sampling loop is part of our proposed method, not an unfair modification applied only to our model.  We found this particularly important in our setting because the posterior $p(x|y)$ over floorplans is highly multimodal, and simple first-order gradient updates were prone to getting stuck in local modes.

---

> ### Author Response · Authors · 2025-11-25
>
> > __Q4. The paper only provides IoU and F1 score as metrics.__
>
> We should have been clearer. The metrics we report in Table 1 of the main paper are averaged results along two axes:
> * average over the whole test dataset;
> * average over 5 samples of plausible floorplans that explain the trajectories (i.e., average of 5 samples from the posterior)
>
> As per your suggestions, we have now evaluated CoGuide on 3 new metrics to assess “correctness” in a distributional sense:
>
> 1. **Floorplan-FID (F-FID):** Since FID is generally computed on InceptionV3 trained on ImageNet, it may not be the right metric to provide meaningful distribution comparison for our floorplan layouts. So, we compute an FID using features from our domain-specific network trained on floorplans. Specifically, instead of retraining another network, we use our trained contrastive floorplan encoder to compute the features across ground truth (all the training data)  and predictions (CFG, CoGuide, CFG+CoGuide). As can be observed in the table below, our proposed CoGuide and CFG+CoGuide models achieve lower FID scores than a CFG-alone baseline.
> 2. **CAS (Contrastive Alignment Score):** Measures (semantic) alignment between generated floorplans and input trajectories in the learned embedding space, with values closer to 1 indicating better alignment. The table demonstrates that our method produces layouts more consistent with trajectories than CFG.
> 3. **SSIM:** We also compute Structural Similarity Index that directly measures pixel-level structural similarity (higher values showing better similarity). We note that combining CFG with CoGuide produces more semantically structured floorplans.
>
> | Method        | F-FID ↓ | SSIM ↑              | CAS ↑              |
> |---------------|:-------:|---------------------|--------------------|
> | **CFG**       | 0.045   | 0.7545 ± 0.1476     | 0.964 ± 0.0116     |
> | **CoGuide**   | 0.040   | 0.7632 ± 0.1175     | 0.975 ± 0.0120     |
> | **CFG+CoGuide** | 0.041 | 0.8498 ± 0.1137     | 0.976 ± 0.0056     |
>
>
> Thank you very much for this comment; I believe this has raised our own confidence in the efficacy and performance of CoGuide.

---

### Author Response · Authors · 2025-11-25
**To all reviewers:**

We thank the reviewers for their thoughtful comments and questions. We took some time to respond to the questions since we were running several new experiments (including real-world results). We have now responded individually to each reviewer below (and have tagged common questions with the reviewer IDs). We have also updated the paper (with blue text, tables, and figures) — together, the revisions have added one new page in the main paper and around 3 new pages in the Appendix section. We look forward to subsequent questions and clarifications; we strongly feel these revisions are sharpening the paper. So thank you again.


Authors,
#22356

---

### Comment · Area_Chair_QGbR · 2025-11-25
**Discussion with Authors**

Dear Reviewers,

The authors have diligently provided responses to your questions and concerns. I request you to please review the authors' responses, acknowledge that you have read them and actively engage with them in further discussion as needed.

This discussion period, with the authors, will end on December 2, 2025 (AoE). However, I request that you not wait until the last minute and actively engage with the authors early.

Best, AC

---

### Author Response · Authors · 2025-12-02
**Final remarks to the AC**

We thank the reviewers, and particularly the AC, for taking on this heavy work-pressure due to the unfortunate turn of events. We are writing a summary comment below, from our point of view.

We think the reviews were fair (including the criticisms). We also think we have addressed most of the criticisms by utilizing the 10 days we had to run experiments. Multiple new results and recommended textual changes have been added to the revised PDF in blue.

__Based on our rebuttals:__

* Reviewer cV1B updated their score from 6 to 8 and also increased their confidence from 3 to 4.
* Reviewers rYah and 7FMZ acknowledged our updates and said they are maintaining “their already positive scores”.
* Reviewer 8553 did not get a chance to respond unfortunately, but a large section of our rebuttal addressed their comments as below.

__New results from the rebuttal phase:__
> __Reviewer 8553 asked about the viability of generalization to one more domain beyond maps.__

We showed generalization to blind audio inverse problems, with new demos in our project webpage (https://coguide.github.io).
We wish 8553 could see them.

> __Reviewers 8553, 7FMZ and cV1B asked about the sim2real gap in our method.__

We performed real-world trajectory measurements inside 2 student apartments using UWB sensors. Our method CoGuide correctly predicted many indoor features in spite of the OOD nature and sparsity of measured trajectories. These new results (added to the paper and website) have further increased our own confidence in the method.

> __Reviewers 8553 and 7FMZ asked about fair baseline comparisons, especially with regards to CFG.__

We performed additional experiments to delineate the benefits of our CoGuide model over CFG. Moreover, we implemented a hybrid CoGuide+CFG model and were able to show superior performance against each individual method (Table 1 in main paper). This table, combined with real-world results (new Fig. 5), serve as a comprehensive comparison against CFG.

In sum, we believe CoGuide offers a promising direction for solving inverse problems with complex forward operators that may be non-differentiable, non-smooth, partially/completely unknown, or even a mixture of all of these scenarios.

Thanks, once again, for a fruitful discussion during the rebuttal phase.

---

### Meta-Review · Area_Chair_xnWK · 2025-12-25

**Summary:**

The authors propose a method for solving spatial inverse problems in situations with a non-differentiable forward operator. All reviewers find the paper to be well-written and the approach interesting and novel. After the initial reviews, which were already positively leaning, a few concerns were mentioned (see below), which have been adequately addressed by the authors in their answers. Since I expect all reviewers would have converged to a positive score, I recommend to accept. Congratulation to the authors for their interesting work!

**Reviewer Concerns:**

*1) Experiments only on a single, synthetic dataset.* The authors addressed this concern by providing an additional set of experiments on reconstructing piano tracks. I think the additional domain clearly strengthens the paper further.

*2) Sim2real gap needs to be analyzed.* The authors provided an analysis of results on real scans.

*3) Additional metrics required.* The authors provided numbers for F-FID, SSIM, and CAS.

*4) Comparison with CFG lacking.* The authors provided an additional comparison with CFG and introduced a version with their method plus CFG, which outperforms both.

*5) Scalability concerns.* Adequately addressed by the audio experiments.

*6) Questions regarding representation choice.* Adequately answered (using the same setup as baselines for fair comparison).

**Reviewer Scores:**

Three reviewers are already positive, where one of them already increased from 6->8.
Only Reviewer 8553 remained negative before the freeze (4). However, since the given concerns were well-addressed by the authors, chances are high that the reviewer would have increased the score to 6, leaving the paper with (8, 6, 6, 6).

---

### Decision · Program_Chairs · 2026-01-26

Accept (Poster)